# Rethinking channel fusion for robust multivariate time series classification under distribution shift

## Abstract

In real-world applications of multivariate time series classification (TSC), distribution shift between training and testing data is common, often leading to degraded out-of-distribution (OOD) performance relative to in-distribution (ID) performance. Existing methods typically improve robustness through the training objective or augmentation, and evaluate models that use early fusion, where channels are jointly processed. However, the impact of fusing channels at later stages remains unclear. We show that later fusion structurally isolates channel-specific shifts, preventing a corrupted channel from contaminating the full feature representation as in early fusion. We evaluate this across four HAR datasets and four MI datasets under both subject-level and sensor corruption distribution shifts. Across HAR datasets, later fusion consistently reduces the ID-OOD gap, and models trained with standard ERM outperform domain generalisation algorithms, often substantially. Later fusion also exhibits strong resilience to sensor corruptions, with late fusion showing near-zero degradation even when half of all channels are corrupted. However, these gains are dataset-dependent: on MI datasets, the ID cost of later fusion outweighs its robustness benefits, while domain generalisation algorithms offer little improvement. We additionally propose a simple ID-based heuristic for selecting fusion strategies. Our findings show that fusion strategy is a critical and underexplored design choice for OOD robustness in multivariate TSC, with effects that can rival those of specialised learning algorithms. The code for this work is available at https://....

## 1 Introduction

Time series classification (TSC) plays an important role in a range of applications, including activity recognition (Zhang et al., 2022), disease diagnosis and monitoring (Oh et al., 2020), predictive maintenance (Carvalho et al., 2019), and climate science (Papagiannopoulou et al., 2017). A key challenge for safe and effective deployment of machine learning models in these applications is robustness to distribution shift: the ability to maintain performance when test data are drawn from a different distribution than the training data. In TSC, such shifts can arise from subject variability, differences in recording equipment, and sensor corruptions. These shifts are often unavoidable in practice; for instance, an activity recognition model on a smartphone needs to generalise to users who were not part of the training data.

We primarily study robustness in the domain generalisation setting: models are trained on a set of source domains, which form the in-distribution (ID) data, and performance is measured on unseen target domains, which are out-of-distribution (OOD). This setting best reflects real-world conditions, where target domain data during training is often unavailable. Gagnon-Audet et al. (2023) showed that models trained with empirical risk minimisation (ERM) (Vapnik, 1991) (i.e. standard neural network training) exhibit substantial differences in their ID and OOD performance across time series benchmarks, highlighting the need for domain generalisation methods in TSC that can narrow this gap. However, they also showed that existing domain generalisation algorithms only offer, at best, marginal improvements in OOD performance over ERM. There remains a clear need for approaches that effectively improve generalisation to unseen domains in TSC.

Prior research in this area has focused on designing learning algorithms, while little attention has been paid to the role of model structure. In this work, we study channel fusion as a component of model structure. In many TSC applications the time series are multivariate, composed of multiple univariate channels that each measure a distinct quantity. At some stage of the model, these channels need to be combined to utilise the information from each one. The fusion stage has been implicitly treated as fixed, with previous works using *early* fusion (Lu et al., 2023; Gagnon-Audet et al., 2023; Ozyurt et al., 2023; He et al., 2023; Mohapatra et al., 2025), in which all channels are jointly processed by a single shared encoder, allowing cross-channel features to be learned. However, alternative fusion strategies are possible: *middle* and *late* fusion, which employ channel-specific encoders and models, respectively, and combine channels at the feature and prediction levels. The impact of fusion strategy on robustness to distribution shift, however, remains unexplored. In this paper, we address this gap.

Our contributions are as follows:

- We identify channel fusion strategy as an overlooked but critical factor in OOD generalisation for multivariate TSC.

- We provide a conceptual analysis of how distribution shifts propagate through early, middle, and late fusion models, showing that early fusion allows a single corrupted channel to affect the full feature representation, while later fusion isolates the perturbation.

- We find, across four human activity recognition (HAR) and four motor imagery (MI) datasets under subject-level distribution shift, that later fusion consistently reduces the ID-OOD gap compared to early fusion, but at the cost of ID performance. When this cost is small, later fusion outperforms domain generalisation algorithms, despite being simpler.

- We find that later fusion is also substantially more robust to sensor corruptions than early fusion, with late fusion showing minimal performance degradation even when half of all channels are corrupted.

- We propose a simple and robust ID-performance-based heuristic for selecting the fusion strategy without access to target data.

- We analyse key design choices within later fusion architectures, showing that (1) encouraging equal channel contributions improves robustness in middle fusion, and (2) uniform weighting performs as well as validation-based weighting in late fusion.

## 2 Related work

### 2.1 Domain generalisation in time series classification

Domain generalisation methods learn from a set of labelled source domains and are evaluated for OOD generalisation on related unseen target domains. A number of general approaches have been proposed, including domain-invariant learning (Ganin et al., 2016; Arjovsky et al., 2020), meta-learning (Li et al., 2018a) and data augmentation (Volpi et al., 2018; Li et al., 2021).

However, empirical studies have cast doubt on the effectiveness of these methods. Gulrajani & Lopez-Paz (2021) compare domain generalisation algorithms on image classification benchmarks and find that none consistently outperform ERM. Proposed reasons for this include: training losses that minimise domain divergence simultaneously worsen ID classification performance (Sener & Koltun, 2022), domain invariance is only achieved across source domains but not beyond (Galstyan et al., 2022), and the number of source domains available during training is too small (Wang et al., 2024).

More recently, domain generalisation methods tailored specifically for time series have been developed, both general-purpose (Lu et al., 2023; Mohapatra et al., 2025) and application-specific (Ragab et al., 2022; Cai et al., 2025). Consistent with findings in vision, Gagnon-Audet et al. (2023) report only marginal gains over ERM across time series tasks.

## 2.2 Channel fusion in time series classification

Middle and late fusion have been used in standard (training and test data are assumed i.i.d.) TSC. Late fusion adapts non-deep univariate time series models to multivariate data by training an ensemble of channel-specific classifiers and combining their predictions (Ruiz et al., 2020; 2021). Early instances of deep learning for TSC use middle fusion, with separate convolutional neural network (CNN) encoders extracting channel-specific features that are concatenated for classification (Zheng et al., 2014). Subsequently, early fusion became standard for state-of-the-art architectures (Fawaz et al., 2019; Foumani et al., 2024).

In OOD generalisation research in TSC, early fusion architectures are used almost exclusively (Wilson et al., 2020; Ragab et al., 2023; Lu et al., 2023; Gagnon-Audet et al., 2023; Ozyurt et al., 2023; He et al., 2023; Liu et al., 2024a; Mohapatra et al., 2025; Liu et al., 2025). In domain adaptation, which aims to adapt a model to a target domain using unlabelled target data, some recent methods have adopted middle fusion (Wang et al., 2023; Kim & Lee, 2025; Ahad et al., 2025). Broadly, these methods take into account the distribution shift in individual channels to aid adaptation. However, these methods are not applicable to domain generalisation as no target data is available during training.

## 2.3 Channel independence in time series forecasting

Recent work in transformer-based multivariate time series forecasting shows that *channel independence* (each channel is processed separately by a shared single-channel model) often outperforms *channel dependence* (channels are processed jointly) (Nie et al., 2023; Zeng et al., 2023; Liu et al., 2024b). Han et al. (2024) characterise this as a capacity-robustness trade-off, where removing inter-channel interactions can improve robustness to distribution shift at the cost of reduced model capacity. However, it remains unclear whether similar trade-offs arise in classification, where both the learning objective and the nature of distribution shifts differ from forecasting.

# 3 Methodology

## 3.1 Problem setup

A multivariate time series with $C$ channels (where $C \geq 2$) and length $L$ is $\mathbf{X} \in \mathbb{R}^{C \times L}$, the $c$th channel of $\mathbf{X}$ is $\mathbf{x}^{(c)} \in \mathbb{R}^L$, and its label is $y \in \{1, \ldots, K\}$ for a $K$-class classification task.

In the domain generalisation setting, $\mathcal{E}_{\text{train}}$ denotes the set of source domains and $\mathcal{E}_{\text{test}}$ the set of target domains, both subsets of $\mathcal{E}_{\text{all}}$. Each domain $d \in \mathcal{E}_{\text{train}} \cup \mathcal{E}_{\text{test}}$ has a labelled dataset $D_d = \{(\mathbf{X}_i, y_i)\}_{i=1}^{n_d}$ with samples drawn i.i.d. from a joint distribution $P_d(\mathbf{X}, y)$. Each distribution is different, such that $P_d \neq P_{d'}$ for all $d \neq d'$. All domains share the same input and label spaces.

Let $f : \mathbb{R}^{C \times L} \to \Delta^{K-1}$ be a classification model that maps an input $\mathbf{X}$ to a vector of class probabilities $(p_1, \ldots, p_K)$, where $p_k = P(y = k \mid \mathbf{X})$. The objective in domain generalisation is to learn $f$ using data from $\mathcal{E}_{\text{train}}$ such that it generalises to unseen domains in $\mathcal{E}_{\text{all}}$. The performance of $f$ on $\mathcal{E}_{\text{test}}$ is used to estimate generalisation to domains in $\mathcal{E}_{\text{all}}$.

## 3.2 OOD generalisation theory

To understand differences in OOD performance between different fusion methods it is necessary to understand what factors determine generalisation to a new domain. Zhao et al. (2018) build on seminal single-source domain adaptation theory (Ben-David et al., 2006; 2010) to develop multi-source domain adaptation theory based on the mixture distribution formed by combining multiple source domains. The datasets from each source domain are combined to form the overall training dataset $D_{\text{train}} = \bigcup_{d \in \mathcal{E}_{\text{train}}} D_d$, which corresponds to drawing samples from the mixture distribution $P_{\text{train}} = \sum_{d \in \mathcal{E}_{\text{train}}} \alpha_d P_d$, where $\alpha_d$ is the mixture weight for domain $d$. We present a population version of the bound and adapt it to deep learning models of the form $f = h \circ g$, following prior work (Johansson et al., 2019; Chuang et al., 2020).

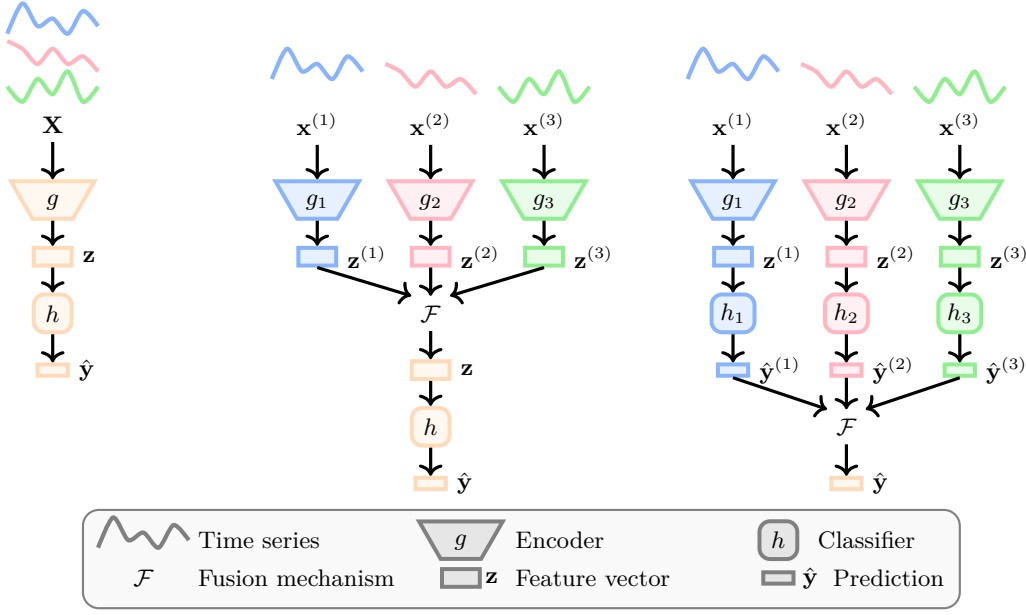

Figure 1: Three general architectures for fusing the channels of multivariate time series: early (left), middle (center), and late (right).

**Theorem 1 (Adapted from Zhao et al. (2018))** *Let $\mathcal{E}_{train}$ and $T$ be the set of source domains and the target domain, respectively. Let $\alpha_d \geq 0$ and $\sum_{d \in \mathcal{E}_{train}} \alpha_d = 1$. With a fixed encoder $g$ and hypothesis class $\mathcal{H}$ of classifiers, the risk in $T$ for all $h \in \mathcal{H}$ is:*

$$R_T(h \circ g) \leq \sum_{d \in \mathcal{E}_{train}} \alpha_d \left[ R_d(h \circ g) + d_{\mathcal{H}} \left( P_d^g(\mathbf{z}), P_T^g(\mathbf{z}) \right) + R_d(h^* \circ g) \right] + R_T(h^* \circ g),$$

*where $R_T = \mathbb{E}_{(\mathbf{X},y) \sim P_T} \ell(f(\mathbf{X}), y)$ and $R_d = \mathbb{E}_{(\mathbf{X},y) \sim P_d} \ell(f(\mathbf{X}), y)$ are the risk in the target domain and source domain $d$ with loss function $\ell$, respectively, $d_{\mathcal{H}}(P_d^g, P_T^g)$ is the $\mathcal{H}$-divergence between the marginal feature distributions induced by $g$, and $h^* = \inf_{h \in \mathcal{H}} \sum_{d \in \mathcal{E}_{train}} \alpha_d R_d(h \circ g) + R_T(h \circ g)$ is the optimal joint hypothesis.*

The bound states that target domain risk is upper-bounded by the $\alpha$-weighted summation of the risk in each source domain, the difference in the feature marginal distributions (domain divergence), and the difference in the label conditional distributions (which determines how well $h^*$ can perform). OOD generalisation research often assumes a covariate shift, where only the marginal distributions differ:

$$P_{\text{train}}(y|\mathbf{X}) = P_T(y|\mathbf{X}), \quad P_{\text{train}}(\mathbf{X}) \neq P_T(\mathbf{X}).$$

Under this assumption, the terms involving $h^*$ become negligible as the optimal classifier is the same across domains. Following this, we focus on source risk and domain divergence as the factors determining OOD performance. We describe the empirical estimates of these terms in Appendix A.

### 3.3 Channel fusion methods

There are three stages at which channel fusion can take place within a deep learning architecture for multivariate time series: early, middle, and late. These are illustrated in Figure 1.

#### 3.3.1 Early fusion

Early fusion consists of fusing channels at the input level, which enables the model to learn cross-channel features that may be more informative than channel-specific features. This is achieved with an encoder $g$ that processes the channels of a multivariate time series together to obtain an $m$-dimensional feature vector

$\mathbf{z} \in \mathbb{R}^m$, which is passed to a classifier $h$ to make a label prediction $\hat{\mathbf{y}}$. The forward pass of an input $\mathbf{X}$ through an early fusion model is:

$$\mathbf{z} = g(\mathbf{X}),$$
$$\hat{\mathbf{y}} = h(\mathbf{z}).$$

A popular choice for $g$ is a 1D CNN. In this model, a single convolutional kernel $\mathbf{K} \in \mathbb{R}^{C \times L_K}$ in the first layer of the model spans all input channels and slides along the time dimension of $\mathbf{X}$. The output $o_t$ of the convolution between $\mathbf{K}$ and $\mathbf{X}$ at time step $t$ is (omitting the bias term):

$$o_t = \sum_{c=1}^{C} \sum_{l=1}^{L_K} x_{t+l}^{(c)} k_l^{(c)},$$

meaning that the output at each time step depends jointly on all input channels. The same applies to other encoder architectures (e.g. transformer, LSTM) but with different formulations.

### 3.3.2   Middle fusion

Middle fusion consists of fusing channels at the level of channel-specific features. While this method also enables cross-channel relationships to be learned, it is between the higher-level features of each channel, rather than the raw inputs. The model is constructed as a set of channel-specific encoders $\{g_c\}_{c=1}^{C}$, and a single classifier $h$. The forward pass is:

$$\mathbf{z}^{(c)} = g_c\left(\mathbf{x}^{(c)}\right),$$
$$\mathbf{z} = \mathcal{F}\left(\mathbf{z}^{(1)}, \ldots, \mathbf{z}^{(C)}\right),$$
$$\hat{\mathbf{y}} = h(\mathbf{z}),$$

where $\mathcal{F}$ is the fusion mechanism, such as summation or concatenation. In this work, we consider the case where $\mathcal{F}$ is a weighted summation of the features from each channel:

$$\mathbf{z} = \sum_{c=1}^{C} w_c \mathbf{z}^{(c)}, \tag{1}$$

with $\sum_{c=1}^{C} w_c = 1$ and $w_c \geq 0$.

In the 1D CNN example, a single convolutional kernel $\mathbf{k}^{(c)} \in \mathbb{R}^{L_K}$ in the first layer of encoder $g_c$ only operates on channel $c$. The output $o_t^{(c)}$ of the convolution between $\mathbf{k}^{(c)}$ and $\mathbf{x}^{(c)}$ at time step $t$ is:

$$o_t^{(c)} = \sum_{l=1}^{L_K} x_{t+l}^{(c)} k_l^{(c)}.$$

There is no operation here across channels, hence the features are channel-specific.

### 3.3.3   Late fusion

Late fusion consists of fusing channels at the level of channel-specific predictions. The model is constructed as an ensemble of channel-specific models $\{f_c = h_c \circ g_c\}_{c=1}^{C}$. The forward pass is:

$$\mathbf{z}^{(c)} = g_c\left(\mathbf{x}^{(c)}\right),$$
$$\hat{\mathbf{y}}^{(c)} = h_c\left(\mathbf{z}^{(c)}\right),$$
$$\hat{\mathbf{y}} = \mathcal{F}\left(\hat{\mathbf{y}}^{(1)}, \ldots, \hat{\mathbf{y}}^{(C)}\right).$$

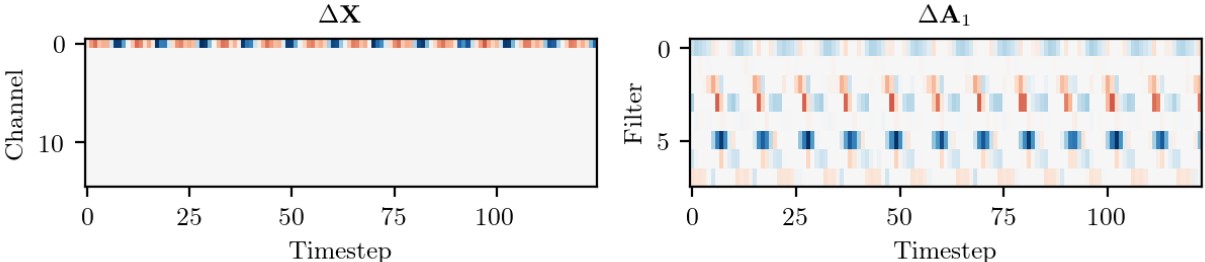

Figure 2: $\Delta\mathbf{X}$ (input difference) and $\Delta\mathbf{A}_1$ (first layer activation difference) for a single instance from `DSADS` under zeroing of the most salient channel. While the perturbation is confined to a single channel in $\Delta\mathbf{X}$, it propagates to most filters in $\Delta\mathbf{A}_1$, illustrating how early fusion allows a single sensor failure to contaminate the full feature representation.

We consider late fusion ensembles where the members are trained independently rather than jointly as this is typical for modern deep ensembles (Jeffares et al., 2023). As a result, this method does not enable cross-channel relationships to be learned. As with middle fusion, we consider the case where the fusion mechanism is a weighted summation:

$$\hat{\mathbf{y}} = \sum_{c=1}^{C} w_c \hat{\mathbf{y}}^{(c)}, \tag{2}$$

with $\sum_{c=1}^{C} w_c = 1, w_c \geq 0$.

It should be noted that, unlike early and middle fusion, late fusion does not produce a single fused feature vector $\mathbf{z}$, meaning there is no feature distribution $P_d^g(\mathbf{z})$, and Theorem 1 does not apply.

### 3.4 Sensitivity to distribution shifts

We now analyse how a distribution shift in a single channel $c$, modelled as a perturbation $\tilde{\mathbf{x}}^{(c)} = \mathbf{x}^{(c)} + \boldsymbol{\delta}$, propagates through each fusion architecture.

**Early fusion.** The perturbation enters the shared encoder alongside the unperturbed channels, and the resulting changes in the model outputs are:

$$\Delta\mathbf{z}_{\text{early}} = \tilde{\mathbf{z}} - \mathbf{z} = g(\tilde{\mathbf{X}}) - g(\mathbf{X}),$$
$$\Delta\hat{\mathbf{y}}_{\text{early}} = h(\tilde{\mathbf{z}}) - h(\mathbf{z}).$$

To visualise the impact that allowing interactions between channels in the encoder can have, we take a 1D CNN trained on the `DSADS` dataset (see Section 4.1), select a random instance from the source validation set, and zero the most important channel (determined with gradient-based saliency) to simulate a sensor failure. We pass both $\mathbf{X}$ and $\tilde{\mathbf{X}}$ through the model and collect the activations after the first layer to obtain $\mathbf{A}_1$ and $\tilde{\mathbf{A}}_1$ respectively. We compute the differences $\Delta\mathbf{X} = \mathbf{X} - \tilde{\mathbf{X}}$ and $\Delta\mathbf{A}_1 = \mathbf{A}_1 - \tilde{\mathbf{A}}_1$, shown in Figure 2. While the perturbation is confined to a single channel in $\Delta\mathbf{X}$, the change propagates to most filters in $\Delta\mathbf{A}_1$, demonstrating how early fusion allows a localised input corruption to contaminate the full feature representation.

**Middle fusion.** The perturbation only enters the encoder $g_c$ and affects the channel-specific feature vector $\mathbf{z}^{(c)}$. Letting $\boldsymbol{\Delta}_{\mathbf{z}}^{(c)} = \tilde{\mathbf{z}}^{(c)} - \mathbf{z}^{(c)}$ denote this change, the changes are:

$$\Delta\mathbf{z}_{\text{middle}} = w_c \boldsymbol{\Delta}_{\mathbf{z}}^{(c)},$$
$$\Delta\hat{\mathbf{y}}_{\text{middle}} = h(\mathbf{z} + w_c \boldsymbol{\Delta}_{\mathbf{z}}^{(c)}) - h(\mathbf{z}).$$

The impact of the perturbation is isolated to $\mathbf{z}^{(c)}$, leaving the other channel-specific features unaffected. The perturbation then enters the fused feature through the weighted sum, and the shared classifier operates on

Table 1: The fusion implementations for middle and late fusion evaluated in this work. For middle fusion, the design choices are equal or learned fusion weights and $\ell_2$-normalisation of channel-specific features. For late fusion, the design choice is equal or validation score-based fusion weights.

| Method | Equal $w_c$ | Learned $w_c$ | Validation $w_c$ | Normalised | Fusion equation |
|---|---|---|---|---|---|
| `middle` | | ✓ | | | $\mathbf{z} = \sum_{c=1}^{C} w_c \mathbf{z}^{(c)}$ |
| `middle-e` | ✓ | | | | $\mathbf{z} = \frac{1}{C} \sum_{c=1}^{C} \mathbf{z}^{(c)}$ |
| `middle-n` | | ✓ | | ✓ | $\mathbf{z} = \sum_{c=1}^{C} w_c \frac{\mathbf{z}^{(c)}}{\|\mathbf{z}^{(c)}\|}$ |
| `middle-e-n` | ✓ | | | ✓ | $\mathbf{z} = \frac{1}{C} \sum_{c=1}^{C} \frac{\mathbf{z}^{(c)}}{\|\mathbf{z}^{(c)}\|}$ |
| `late` | ✓ | | | | $\hat{\mathbf{y}} = \frac{1}{C} \sum_{c=1}^{C} \hat{\mathbf{y}}^{(c)}$ |
| `late-v` | | | ✓ | | $\hat{\mathbf{y}} = \sum_{c=1}^{C} w_c \hat{\mathbf{y}}^{(c)}$ |

the result. While the impact on $\Delta\mathbf{z}_{\text{middle}}$ is linear, the impact on $\Delta\hat{\mathbf{y}}_{\text{middle}}$ may not be, depending on the classifier $h$. When $h$ is linear, the change in the prediction simplifies to:

$$\Delta\hat{\mathbf{y}}_{\text{middle}} = h(\mathbf{z} + w_c \mathbf{\Delta}_{\mathbf{z}}^{(c)}) - h(\mathbf{z}) = w_c h(\mathbf{\Delta}_{\mathbf{z}}^{(c)}),$$

and the effect of the perturbation remains linear and scaled by the fusion weight $w_c$. However, if $h$ is non-linear, this equality does not hold and the perturbation can interact with the full fused feature through the non-linearities of $h$.

**Late fusion.** The perturbation affects the channel-specific encoder $g_c$ and classifier $h_c$, affecting the channel-specific prediction $\hat{\mathbf{y}}^{(c)}$. Letting $\mathbf{\Delta}_{\hat{\mathbf{y}}}^{(c)} = \tilde{\mathbf{y}}^{(c)} - \hat{\mathbf{y}}^{(c)}$ denote this change, this yields:

$$\Delta\hat{\mathbf{y}}_{\text{late}} = w_c \mathbf{\Delta}_{\hat{\mathbf{y}}}^{(c)}.$$

In late fusion, the impact is isolated similarly to middle fusion, but all the way through to the prediction stage. Only $\hat{\mathbf{y}}^{(c)}$ is affected, and the perturbation enters the final predicted output through the weighted contribution of channel $c$.

**Shifts in multiple channels.** A distribution shift may occur in multiple channels rather than just one. For example, multiple sensors can become noisy or saturated, or all channels could shift under a new subject. Because channel independence is maintained prior to fusion, the overall effect in middle and late fusion is the weighted sum of the individual channel-specific effects. Let $\mathcal{S}$ denote the set of shifted channels. The changes are:

$$\Delta\mathbf{z}_{\text{middle}} = \sum_{c \in \mathcal{S}} w_c \mathbf{\Delta}_{\mathbf{z}}^{(c)}, \quad \Delta\hat{\mathbf{y}}_{\text{late}} = \sum_{c \in \mathcal{S}} w_c \mathbf{\Delta}_{\hat{\mathbf{y}}}^{(c)}.$$

In early fusion, simultaneous shifts can interact within the shared non-linear layers and the effect on $\Delta\mathbf{z}_{\text{early}}$ cannot be decomposed as individual channel-specific contributions.

### 3.5 Design choices for middle and late fusion

Following from the previous analysis, we consider the role of fusion weights and feature magnitudes in later fusion. To isolate their effects, we evaluate several implementations of middle and late fusion, summarised in Table 1.

**Fusion weights in middle fusion.** Because the effect of a shift in channel $c$ on $\Delta\mathbf{z}_{\text{middle}}$ scales with $w_c$, fusion weights can amplify or attenuate their impact, and therefore play an important role in robustness. As middle fusion models are trained jointly, these weights can be learned during training. They may be fixed across inputs or input-dependent via an attention mechanism. In this work, we focus on the former.

**Feature magnitudes in middle fusion.** Beyond the explicit fusion weights, a channel encoder can implicitly inflate or suppress its contribution to $\mathbf{z}$ by adjusting the magnitude of $\mathbf{z}^{(c)}$ relative to other

channels. This provides an alternative route through which a shift in channel $c$ can affect the fused feature, independently of $w_c$. To control for this, we normalise each channel-specific feature vector to have unit $\ell_2$ norm prior to fusion.

**Fusion weights in late fusion.** Fusion weights in late fusion cannot be learned during training when the channel-specific models are trained independently. Instead, they must be specified heuristically. Common choices include uniform weighting and weighting by validation performance (Large et al., 2019); we consider both. For the latter, let $m_c$ denote a performance metric computed on the validation set for channel $c$. Where a higher value for $m_c$ is better (e.g. accuracy), we set the weight for channel $c$ as:

$$w_c = \frac{\exp(m_c/\tau)}{\sum_{i=1}^{C} \exp(m_i/\tau)}, \tag{3}$$

where $\tau$ is the temperature hyperparameter. Smaller values of $\tau$ put more weight on the better performing channels, with a single channel receiving a weight of 1 as $\tau \to 0$. Larger values of $\tau$ spread the weights more evenly across channels, and uniform weights are recovered as $\tau \to \infty$. As target data is unavailable, $\tau$ cannot be tuned on the target domain. We therefore analyse `late-v` across different values of $\tau$.

### 3.6 A heuristic for fusion selection

Selecting which fusion method to use is difficult in the domain generalisation setting because of the lack of target domain data. This is an instance of underspecification (D'Amour et al., 2022): the ID validation signal does not uniquely determine OOD behaviour, so two fusion methods with equivalent ID performance may generalise very differently.

As described in Section 3.2, the two factors that influence OOD performance are ID performance and domain divergence. As this framework applies only to early and middle fusion, we more broadly characterise OOD performance in terms of ID performance and robustness to distribution shift. Computing the latter requires data from the target domain, so we are restricted to using the ID performance for selection.

Our experiments with `early`, `middle`, and `late` reveal consistent observations that motivate a simple heuristic for selecting between them. We find that ID performance generally follows the order: `early` $>$ `middle` $>$ `late` (Section 4.2), while robustness (defined here as the ID-OOD generalisation gap) follows the reverse order: `late` $>$ `middle` $>$ `early` (Section 4.3 and 4.4). This means that selecting based solely on ID performance would consistently favour early fusion. We propose a selection method that defaults to the most robust architecture (`late`), and switches if the ID performance of one of the other fusion methods exceeds that of `late` by more than a tolerance $\delta$.

Let $m_{\text{early}}$, $m_{\text{middle}}$, and $m_{\text{late}}$ denote a performance metric computed on the ID validation set for each fusion method, where a higher value indicates better performance (for example accuracy or negative cross-entropy). We apply the following hierarchical decision rule:

$$f^* = \begin{cases} \texttt{late} & \text{if } m_{\text{late}} \geq \max(m_{\text{middle}}, m_{\text{early}}) - \delta, \\ \texttt{middle} & \text{if } m_{\text{middle}} \geq m_{\text{early}} - \delta, \\ \texttt{early} & \text{otherwise,} \end{cases}$$

where $\delta \geq 0$ is a tolerance hyperparameter that dictates how much degradation in ID performance is acceptable in exchange for increased robustness. For bounded metrics such as accuracy or macro F1, $\delta = 0.05$ corresponds to accepting a decrease of 5 percentage points. For unbounded loss-based metrics, a simple normalisation (for example, dividing cross-entropy by $\log K$) can be applied to ensure $\delta$ is comparable across datasets.

## 4 Experiments

### 4.1 Experimental set-up

**Datasets.** We perform experiments on four HAR datasets: `DSADS` (Altun et al., 2010), `MHEALTH` (Banos et al., 2014), `PAMAP` (Reiss & Stricker, 2012), `WISDM` (Kwapisz et al., 2011), and four MI datasets: `BNCI-1`

Table 2: Summary statistics of the datasets used in our experiments. $N$ is the number of subjects, $K$ is the number of classes, $C$ is the number of channels, $f_s$ is the sampling frequency in Hz, and $L$ is the sequence length in timesteps.

| Name | Task | $N$ | $K$ | $C$ | $f_s$ | $L$ | No. samples |
|---|---|---|---|---|---|---|---|
| DSADS | HAR | 8 | 19 | 15 | 25 | 125 | 9120 |
| MHEALTH | HAR | 10 | 12 | 23 | 50 | 100 | 3317 |
| PAMAP | HAR | 9 | 18 | 30 | 50 | 150 | 3415 |
| WISDM | HAR | 36 | 6 | 3 | 20 | 128 | 8198 |
| BNCI-1 | MI | 9 | 4 | 22 | 100 | 400 | 5184 |
| BNCI-2 | MI | 14 | 2 | 15 | 100 | 500 | 2240 |
| BNCI-4 | MI | 9 | 2 | 3 | 100 | 750 | 6520 |
| ZHOU | MI | 4 | 3 | 14 | 100 | 500 | 1800 |

(Tangermann et al., 2012), `BNCI-2` (Steyrl et al., 2016), `BNCI-4` (Leeb et al., 2007), and `ZHOU` (Zhou et al., 2016). Summary statistics are shown in Table 2, and further details on each dataset and preprocessing steps are provided in Appendix B.

**Evaluation protocol.** For all datasets, each subject is defined as a domain. Let $\mathcal{E} = \{d_1, \ldots, d_N\}$ denote the set of all domains. We use leave-one-domain-out cross-validation: in each fold, one domain is used as the unseen OOD target domain $\mathcal{E}_{\text{test}} = \{d_i\}$, and the remaining domains form the set of source domains $\mathcal{E}_{\text{train}} = \mathcal{E} \setminus \mathcal{E}_{\text{test}}$ for training. ID metrics are computed on the validation set of each source domain, and OOD metrics are computed on the target domain.

**Metrics.** We use the macro F1 score to measure classification performance. Let $m_d^{\text{OOD}}$ and $m_d^{\text{ID}}$ denote the OOD and ID macro F1 scores when $d \in \mathcal{E}$ is the held-out target domain, respectively. We report the following metrics:

- Average ID performance: $m_{\text{avg}}^{\text{ID}} = \frac{1}{|\mathcal{E}|} \sum_{d \in \mathcal{E}} m_d^{\text{ID}}$

- Average OOD performance: $m_{\text{avg}}^{\text{OOD}} = \frac{1}{|\mathcal{E}|} \sum_{d \in \mathcal{E}} m_d^{\text{OOD}}$

- Worst-group performance (Sagawa et al., 2020): $m_{\text{worst}}^{\text{OOD}} = \min_{d \in \mathcal{E}} m_d^{\text{OOD}}$

- Average generalisation gap (Sener & Koltun, 2022): $\text{gap}_{\text{avg}} = \frac{1}{|\mathcal{E}|} \sum_{d \in \mathcal{E}} \left( m_d^{\text{ID}} - m_d^{\text{OOD}} \right)$

**Algorithms.** We use ERM to train our implementations of early, middle, and late fusion. We compare with a time series-specific domain generalisation method: `PhASER` (Mohapatra et al., 2025), and three popular general domain generalisation algorithms: maximum mean discrepancy (`MMD`) (Li et al., 2018b), variance risk extrapolation (`VREx`) (Krueger et al., 2021), and nuclear norm regularisation (`NNR`) (Shi et al., 2024). Details of these algorithms are provided in Appendix C. Domain generalisation algorithms typically require tuning one or more hyperparameters, which is challenging given the lack of target domain data. We select hyperparameters based on OOD performance, which is an optimistic view of their performance, but the aim here is to compare them when they perform well.

**Models.** We use a 1D CNN as the encoder unless stated otherwise. Details on architectures, training, and hyperparameters are provided in Appendix D.

All results are averaged over five runs with different random seeds, and reported with $\pm$ one standard deviation across runs. The complete set of results for all methods are provided in Appendix E.

## 4.2 How does fusion impact ID performance?

We first examine ID performance as one of the determining factors of OOD performance. Figure 3 shows the results for `early`, `middle`, and `late`.

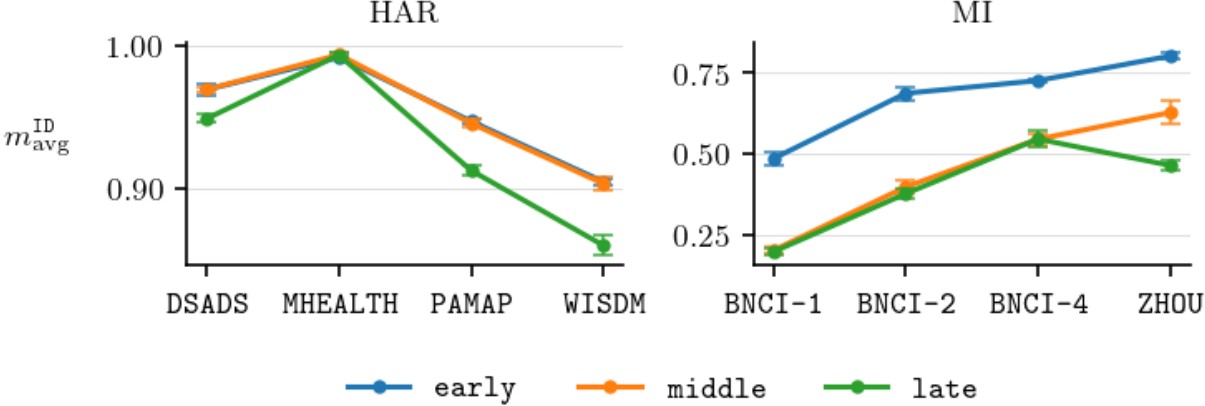

Figure 3: Average ID performance $m^{\text{ID}}_{\text{avg}}$ across datasets for early, middle, and late.

Table 3: The average pairwise correlation between channels $\bar{\rho}$ across datasets.

|  | DSADS | MHEALTH | PAMAP | WISDM | BNCI1 | BNCI2 | BNCI4 | ZHOU |
|---|---|---|---|---|---|---|---|---|
| $\bar{\rho}$ | 0.261 | 0.227 | 0.191 | 0.269 | 0.763 | 0.768 | 0.575 | 0.552 |

The different fusion methods permit different degrees of cross-channel interaction: early learns non-linear cross-channel relationships at the input level, middle learns additive relationships at the feature level, and late prohibits cross-channel learning. Allowing more complex channel interactions increases the capacity of the model to fit the training distribution, which is reflected in the ID results: while early and middle perform similarly on HAR, early significantly outperforms middle on MI, and late tends to perform slightly worse than middle on both tasks.

More capacity may be needed for the MI datasets because EEG signals have a lower signal-to-noise ratio than IMU signals, making it more difficult to learn class-discriminative features from individual channels. While individual channels may be weakly discriminative, useful information may emerge from relationships between channels, which early is best able to exploit. To understand the cross-channel structure of each dataset, we measure the average absolute Pearson correlation between pairs of channels:

$$\bar{\rho} = \frac{1}{|D_{\text{train}}|} \sum_{\mathbf{X} \in D_{\text{train}}} \frac{2}{C(C-1)} \sum_{i=1}^{C} \sum_{j=i+1}^{C} |\rho_{ij}(\mathbf{X})|,$$

where $\rho_{ij}(\mathbf{X})$ is the Pearson correlation between channels $i$ and $j$ for instance $\mathbf{X}$. The MI datasets exhibit substantially higher cross-channel correlation than the HAR datasets (Table 3), suggesting that relationships between channels may contain useful discriminative information, helping explain why early benefits more on MI.

### 4.3 How does middle fusion impact domain divergence?

We now look at domain divergence as the other factor determining OOD performance. We restrict this analysis to early and middle for two reasons. First, as discussed in Section 3.3.3, Theorem 1 does not apply to late fusion. Second, statistical distances are not scale-invariant, making comparisons with the normalised middle fusion variants unfair.

We report domain divergence using the Wasserstein distance computed with POT (Flamary et al., 2021), and average over target domains: $\text{Div}_{\text{avg}} = \frac{1}{|\mathcal{E}|} \sum_{d \in \mathcal{E}} \text{Div}_d$. We also report the average feature magnitude

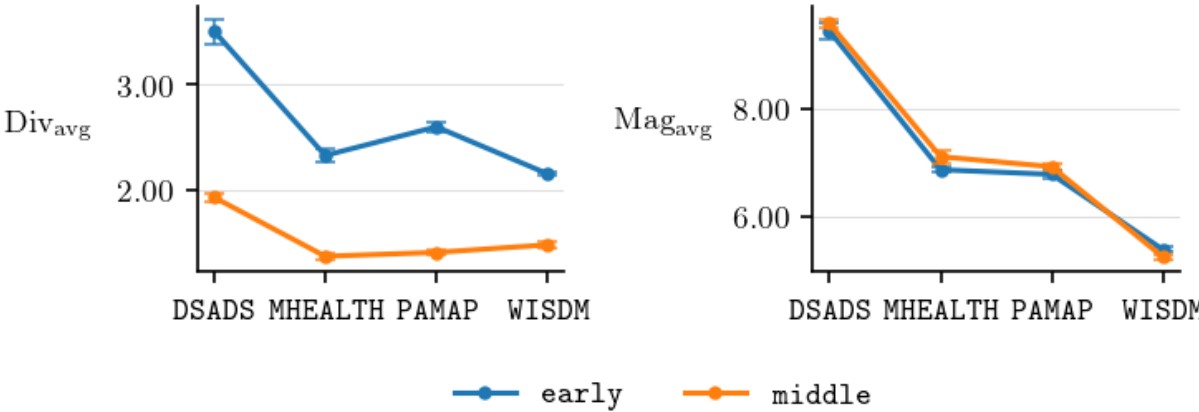

Figure 4: Average domain divergence $\text{Div}_{\text{avg}}$ and feature magnitude $\text{Mag}_{\text{avg}}$ across the HAR datasets for `early` and `middle`.

as $\text{Mag}_{\text{avg}} = \frac{1}{|\mathcal{E}|} \sum_{d \in \mathcal{E}} \text{Mag}_d$, with $\text{Mag}_d = \frac{1}{|D_{\text{train}}|} \sum_{\mathbf{x} \in D_{\text{train}}} \|\mathbf{z}\|_2$. We restrict this analysis to the HAR datasets as the feature magnitudes of `early` and `middle` are not comparable on the MI datasets.

The results are shown in Figure 4. The feature magnitudes are similar, enabling a fair comparison of domain divergence. Across all datasets, `middle` yields smaller domain divergence than `early`. This indicates better alignment between source and target domain features, which suggests `middle` is more robust to distribution shift than `early`. Whether this translates to better OOD performance is examined in the following section.

### 4.4 How does fusion impact OOD performance?

We have seen how ID performance and domain divergence are impacted by different fusion methods, and we now examine the resulting OOD performance. The three metrics we consider are shown in Figure 5 for `early`, `middle`, `late`, and `best-dg`, which is the best performing domain generalisation algorithm on each dataset. Full results are provided in Table 4.

**Later fusion improves OOD performance when ID degradation is small.** On the HAR datasets, where the ID performances between fusion methods are similar, `middle` and `late` substantially outperform `early` on all three metrics. In contrast, on the MI datasets, `early` achieves the best OOD performances as the ID performance cost of later fusion is too large to overcome. Overall, these results reveal a consistent pattern: *later fusion is an effective means of improving OOD performance when the accompanying reduction in ID performance is small.*

**Later fusion helps more than domain generalisation algorithms for HAR.** On all the HAR datasets besides `WISDM`, later fusion (particularly `late`) outperforms `best-dg`, suggesting it can be a more effective method than domain generalisation algorithms on datasets where ID performance is mostly maintained. `PhASER` slightly outperforms `middle` and `late` on `WISDM`. This may be a consequence of `WISDM` having far fewer channels than the other HAR datasets (3 versus 15, 23, 30), which could limit the benefit of channel isolation in later fusion.

**Robustness interventions do not help on MI.** `best-dg` does not suffer the same OOD degradation on the MI datasets as `middle` and `late`, because its ID performance is better preserved (Table 4). An explanation for this is that, with smaller algorithm-specific penalties, these algorithms behave more similarly to standard ERM (Appendix C), so performance is more similar to `early`. However, they also do not provide any meaningful improvement over `early`. This suggests that for MI, ID performance is the key quantity and attempts to improve robustness do not contribute to improving OOD performance.

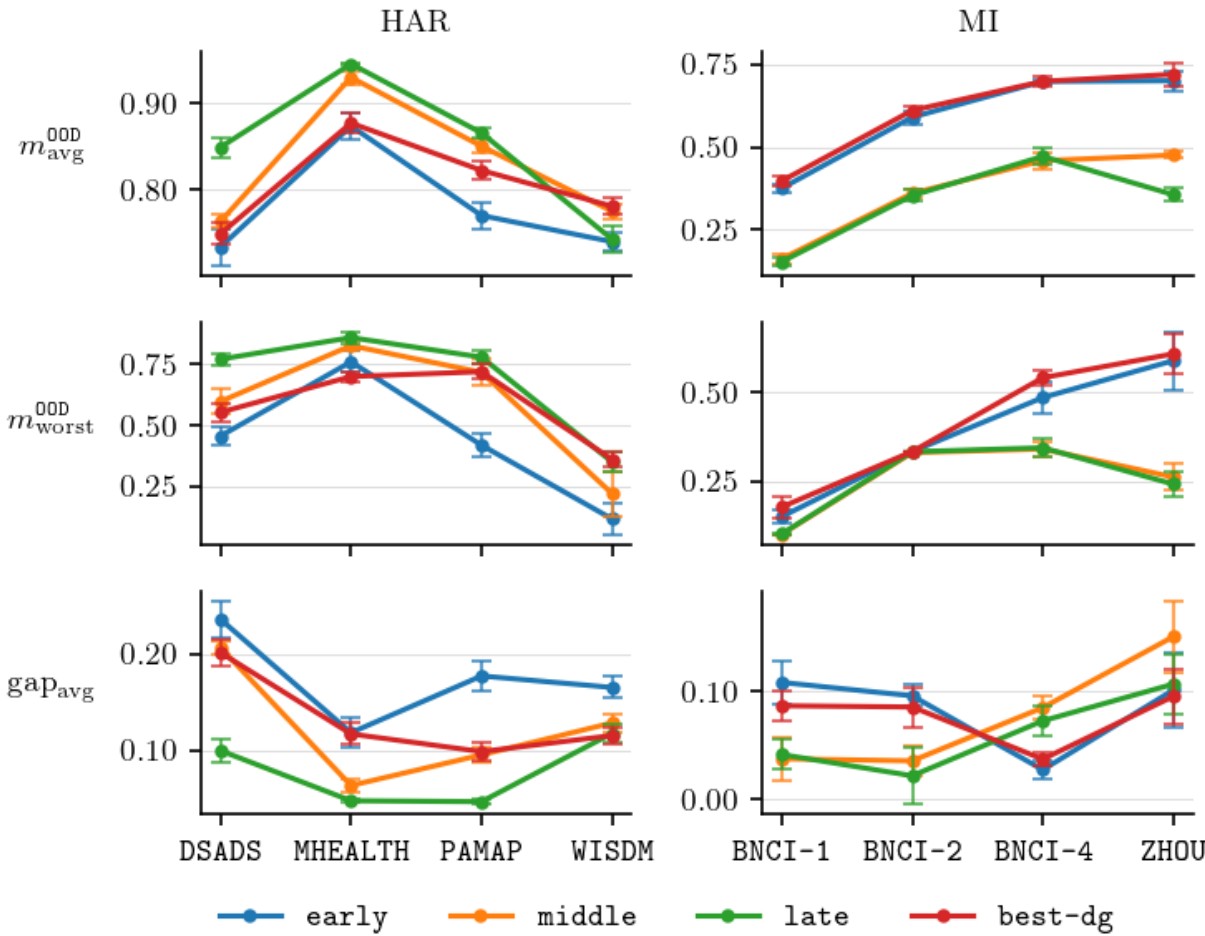

Figure 5: Average OOD performance $m_{\text{avg}}^{\text{OOD}}$, worst-group performance $m_{\text{worst}}^{\text{OOD}}$, and generalisation gap $\text{gap}_{\text{avg}}$ across datasets for `early`, `middle`, `late` and `best-dg` (the best performing domain generalisation algorithm on each dataset).

**Late fusion outperforms middle fusion for HAR.** On the HAR datasets, `late` outperforms `middle` on most metrics. In the following section, we investigate the middle fusion variants to better understand the factors behind this difference.

### 4.4.1 Encoder types and model size

**Encoder architectures.** In Appendix E.1, we show the OOD performance for `early`, `middle`, and `late` with two different encoder architectures that are popular in TSC: CNN-LSTM and transformer. We observe the same general trends across all evaluation metrics, indicating that our findings are not specific to a 1D CNN encoder.

**Parameter count.** When each channel-specific encoder in middle and late fusion has the same structure as the early fusion encoder (aside from the number of input channels), the model size scales with the number of channels. In Appendix E.2, we study the effect of model size by increasing the width (number of filters, $n_f$) of each layer of the 1D CNN encoder for `early`, in order to compare performance under more comparable parameter counts. The results show that `middle` and `late` with $n_f = 8$ still outperform `early` with $n_f = 64$ on the HAR datasets, despite having fewer parameters. This confirms that the improved performance on these datasets is not simply a consequence of larger parameter count.

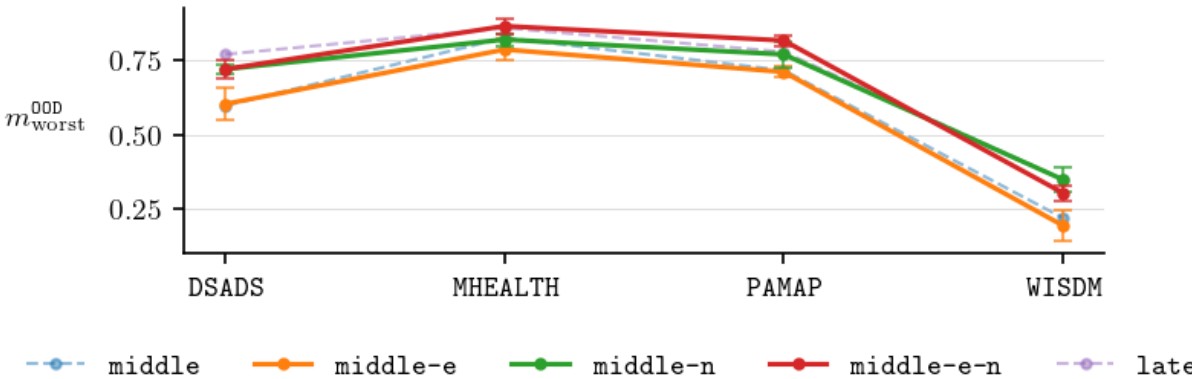

Figure 6: Worst-group OOD performance $m_{\text{worst}}^{\text{OOD}}$ across HAR datasets for the middle fusion variants, with `middle` and `late` for reference.

### 4.5 Design choices in later fusion

In this experiment, we consider the variants of middle and late fusion (Table 1) to study how fusion weights and feature magnitude impact OOD performance. We only consider the HAR datasets as we have seen that later fusion is less suitable for MI.

#### 4.5.1 Middle fusion variants

Figure 6 shows the worst-group OOD performance for the middle fusion variants and `late`. Comparing `middle-e` and `middle-n`, normalising channel-specific features has a more positive impact on OOD performance than equal fusion weights. While equal fusion weights alone do not improve performance over `middle`, using them on top of normalised feature vectors (i.e. `middle-e-n`) does help on `MHEALTH` and `PAMAP`. These results show that encouraging equal channel contributions helps to improve robustness in middle fusion, as it prevents over-weighting channels that experience more severe distribution shift.

#### 4.5.2 Fusion weights in late fusion

We now analyse `late-v`, which uses the validation performance of each channel-specific model to set the fusion weights as in Equation 3. The top row of Figure 7 shows the OOD performance of `late-v` for values of $\tau$ spaced logarithmically in $[0.01, 10]$. For reference, the average value of $\tau$ at which the fusion weights are close to uniform is shown, defined as: $\bar{\tau}^* = \frac{1}{|\mathcal{E}|} \sum_{d \in \mathcal{E}} \tau_d^*$, where $\tau_d^* = \min\left\{\tau : \max_c \left|w_c^d(\tau) - \frac{1}{C}\right| < \epsilon\right\}$ and $\epsilon = 0.01$. The bottom row shows the OOD performance for each channel-specific model.

As $\tau \to 0$, OOD performance approaches that of the model with the best ID performance, which is far worse than using all models with uniform weighting (except for `WISDM`). Performance improves monotonically from $\tau = 0.01$ to approximately $\tau = 0.1$ as the weights become more evenly distributed across models. Performance stabilises at $\tau > 0.1$, showing that uniform weights perform comparably to the best achievable weighting, and that little is gained by differentiating between channels based on their validation performance. On `WISDM`, there is a small drop in OOD performance as $\tau$ is increased towards uniform weights, suggesting a weak benefit to validation-based weighting on this dataset.

### 4.6 Fusion selection method

We now examine how the fusion selection method performs for selecting between `early`, `middle`, and `late`. Figure 8 shows the OOD performance and the proportion of target domains selected for each fusion method across values of the tolerance hyperparameter $\delta$.

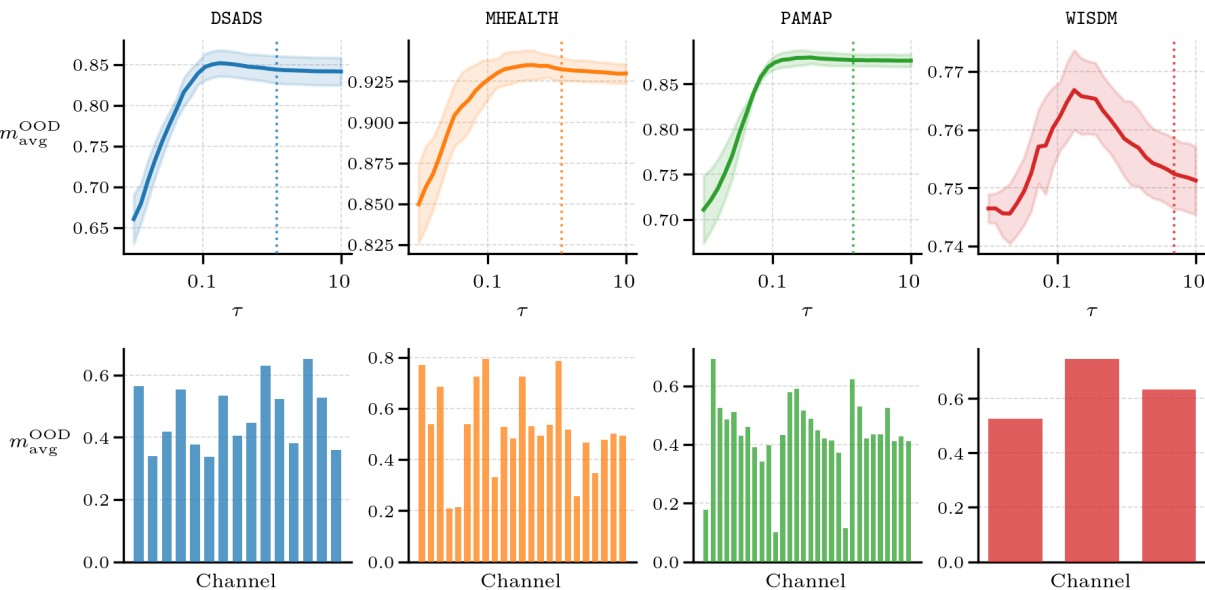

Figure 7: Top row: average OOD performance $m_{\text{avg}}^{\text{OOD}}$ across a range of values for $\tau$ in the channel weighting of `late-v`. The dotted line represents $\bar{\tau}^*$ for each dataset. Bottom row: average OOD performance of each channel-specific model $f_c$.

On the HAR datasets, where `early` and `middle` achieve similar ID performance, selection favours `middle` at smaller values of $\delta$. `middle` is selected over `early` because of the selection rule hierarchy, and over `late` as there is less tolerance for ID degradation at smaller values of $\delta$. As $\delta$ increases, selection favours `late` over `middle` as the ID degradation becomes smaller than the allowed tolerance. On the MI datasets, where `middle` and `late` fusion exhibit large ID performance degradation relative to `early`, `early` is primarily selected, with `middle` and `late` selected at larger values of $\delta$.

Across all datasets, the selection method matches or exceeds the best fixed fusion method over some range of $\delta$; we find that $\delta \in [0.05, 0.1]$ works well across the different datasets, indicating the method is not sensitive to precise tuning of $\delta$.

### 4.7  Robustness to sensor corruptions

The previous experiments investigate robustness under subject-level distribution shifts. We now consider sensor corruptions as a different type of shift to further understand the robustness of each fusion method. These shift types generally differ in two important ways. First, sensor corruptions might be localised to a subset of channels, while subject-level shifts are more likely to impact all channels. Second, channels under subject-level shifts generally remain informative with respect to the class label, while certain sensor corruptions can render a channel uninformative.

**Experiment setup.** To isolate the effect of channel corruption from subject-level variation, we use the source validation set as the test data. We compute the importance of each channel for a trained `early` model using gradient-based saliency, and then apply sensor corruptions to the top-$k$, bottom-$k$, and random-$k$ most important channels of the test data. The maximum value of $k$ is set to $\lfloor C/2 \rfloor$. Three types of shifts are used, following Ahad et al. (2025): zero (channels are set to all zeros), saturate (channels are set to a value drawn from $U(1, 3)$), and noise (additive Gaussian noise from $\mathcal{N}(0, \sigma)$ with $\sigma \sim U(1, 3)$).

The results for this experiment on `DSADS` are shown in Figure 9. For all methods, performance is closely tied to both the importance and number of corrupted channels. It drops as more channels are corrupted, and for a fixed number of corrupted channels, the drop is generally largest for top-$k$, followed by random-$k$, and then bottom-$k$. The ordering of the performances of each method is largely consistent with that of the

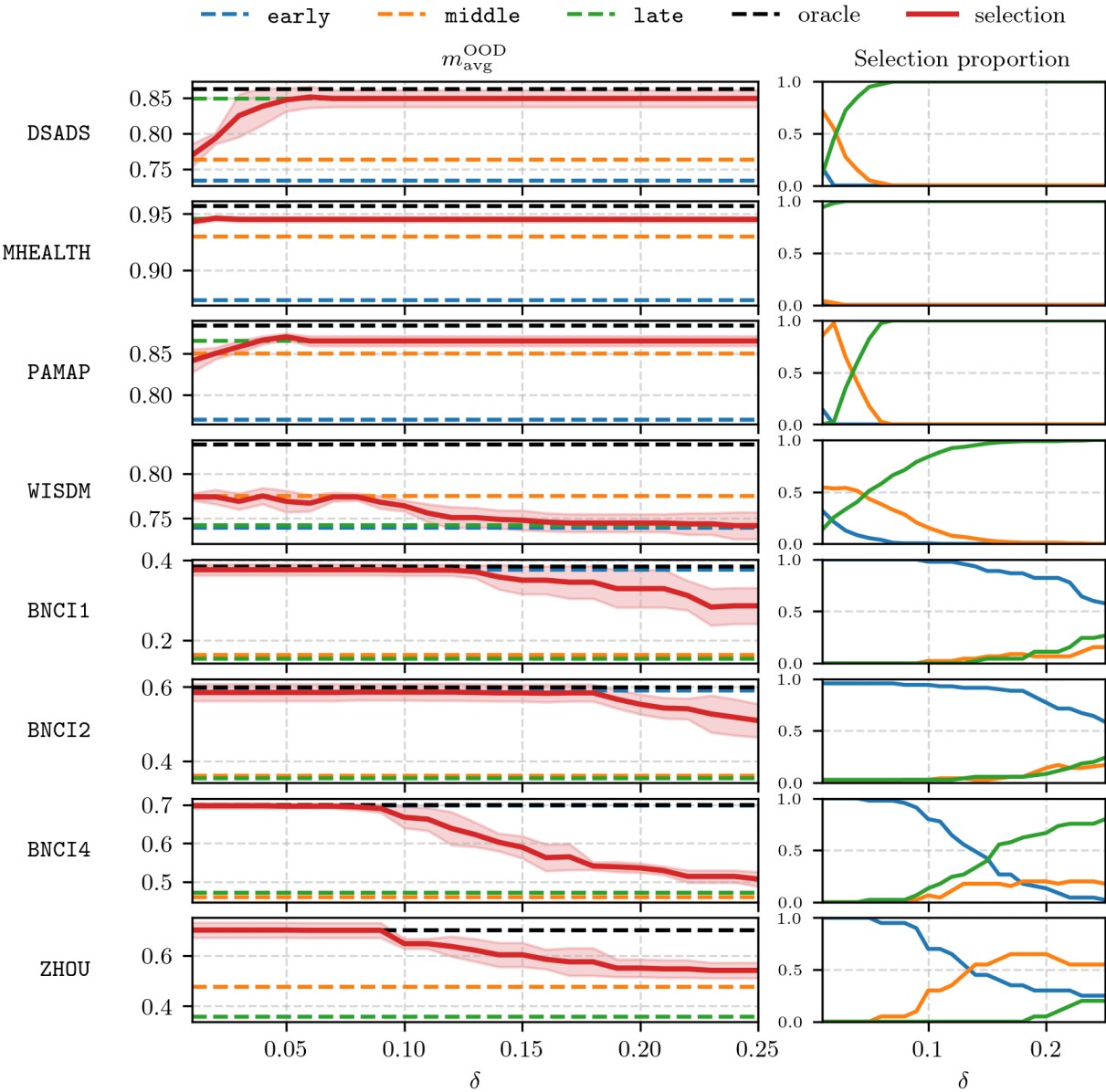

Figure 8: Left: Average OOD performance $m_{\mathrm{avg}}^{\mathrm{OOD}}$ across values of the tolerance hyperparameter $\delta$ with the fusion selection method. The oracle performance is the performance when the best fusion method on each individual target domain is selected. Right: the proportion of target domains for each fusion method across values of $\delta$.

subject-level shift experiments; across both importance and number of shifts, the ordering follows: `early` < `middle` < `middle-e-n` < `late`. In particular, `late` shows remarkable resilience to sensor corruptions with minimal performance degradation even when 7 out of 15 channels are corrupted. `middle-e-n` largely outperforms `middle`, reinforcing the earlier finding that encouraging uniform channel contributions improves robustness.

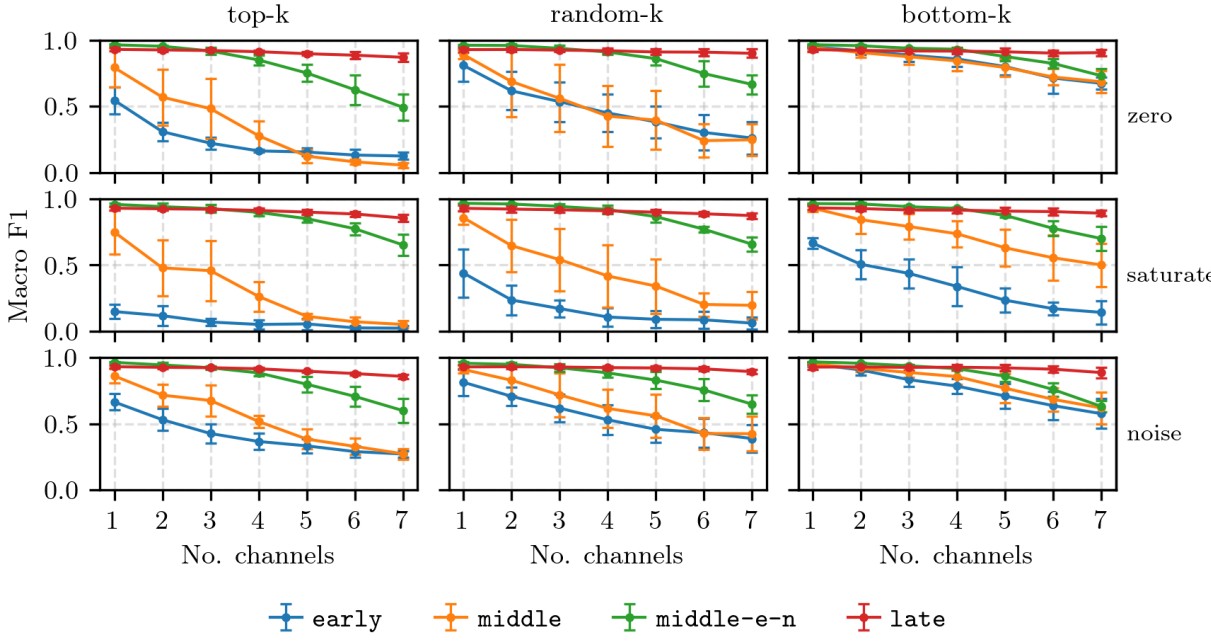

Figure 9: Macro F1 score on `DSADS` under three types of sensor corruption (right side): zero, saturate, and noise. Corruptions are applied to the top-$k$, random-$k$, and bottom-$k$ most important channels.

## 5 Conclusion

In this work, we show that channel fusion is a critical design choice for OOD generalisation in multivariate TSC. Later fusion models consistently outperform early fusion and domain generalisation algorithms on HAR datasets under both subject-level and sensor corruption distribution shifts. However, these gains are dataset-dependent: on MI, later fusion degrades performance relative to early fusion, and domain generalisation algorithms offer no meaningful improvement. We therefore introduce a practical approach for selecting the fusion strategy without access to target data.

The field's implicit reliance on early fusion risks overstating the effectiveness of domain generalisation algorithms while also obscuring the role of model structure. This work challenges this practice, showing that structure can matter as much as the choice of learning algorithm.

This work has several limitations that open up directions for future research. While we evaluate middle and late fusion with weighted summation, alternative fusion mechanisms, such as concatenation or attention-based approaches, may exhibit different behaviour. Since our experiments focus on the TSC tasks of HAR and MI, it remains unclear whether these findings extend to other tasks. Additionally, we consider only subject-level and sensor corruption shifts; other forms, such as device-level shifts, remain unexplored. Finally, although later fusion is algorithm-agnostic, our study pairs it exclusively with ERM; combining it with domain generalisation algorithms may yield further improvements.

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

## A Empirical estimates of source risk and domain divergence

**Source / ID risk.** This term captures how well the model classifies unseen data drawn from $P_{\text{train}}$. We use the "training-domain validation set" strategy (Gulrajani & Lopez-Paz, 2021) to estimate source risk. The dataset for each source domain $d$ is split into training and validation subsets $D_d^{\text{train}}$ and $D_d^{\text{val}}$. The empirical risk is computed on each validation set:

$$\hat{R}_{\mathcal{E}_{\text{train}}} = \sum_{d \in \mathcal{E}_{\text{train}}} \alpha_d \hat{R}_d(f), \quad \hat{R}_d(f) = \frac{1}{|D_d^{\text{val}}|} \sum_{(\mathbf{X}, y) \in D_d^{\text{val}}} \ell(f(\mathbf{X}), y).$$

**Domain divergence.** This captures the alignment between source and target feature distributions, reflecting the sensitivity of the encoder to distribution shift. The bound specifies the divergence between feature marginals using the $\mathcal{H}$-divergence, which can be estimated via a classifier two-sample test (Ben-David et al., 2006; Ganin et al., 2016). However, training a classifier is computationally expensive and statistical distances are often used for this purpose in practice (Zhao et al., 2019), particularly the maximum mean discrepancy (MMD) (Long et al., 2015; Liu & Xue, 2021) and the Wasserstein distance (Courty et al., 2017; Shen et al., 2018). We estimate the domain divergence for target domain $T$ as:

$$\text{Div}_T = \sum_{d \in \mathcal{E}_{\text{train}}} \alpha_d \, \text{dist}\left(\hat{P}_d^g(\mathbf{z}), \hat{P}_T^g(\mathbf{z})\right),$$

where $\hat{P}_d^g(\mathbf{z}) = \left\{ \mathbf{z}_i = g(\mathbf{X}_i) \mid \mathbf{X}_i \in D_d \right\}$ is the empirical feature distribution for domain $d$, and $\text{dist}(\cdot, \cdot)$ is a statistical distance.

**Mixture weights.** When combining the datasets from each source domain into one, the empirical mixture weights are proportional to the size of the dataset: $\alpha_d = \frac{|D_d|}{|D_{\text{train}}|}$.

# B Datasets

In this section, we provide details for the benchmark datasets and the preprocessing steps used in our experiments.

**Preprocessing.** Some datasets are provided pre-segmented into input-labels pairs. For datasets that are a single continuous recording, we use a sliding window approach to generate input-label pairs. A non-overlapping window of fixed size is passed along the time series to create segments at fixed intervals. A window is retained only if all the time steps within the window share the same label. After windowing, we only retain the classes that are common among all domains. Each window is normalised independently to have zero mean and unit standard deviation. Finally, each domain dataset is split into training and validation sets with a 70%/30% split, stratified by class label.

## B.1 Human activity recognition datasets

**DSADS** (Altun et al., 2010). This dataset contains data from 8 subjects for the task of human activity recognition, with 19 classes. The data was recorded with a 3-axis accelerometer, 3-axis gyroscope, and 3-axis magnetometer device at five different locations on the body (total 45 channels), with a sampling frequency of 25 Hz. The dataset is already segmented into windows of 125 samples (5 seconds). To ease computation, we use the accelerometer channels only (total 15 channels).

**MHEALTH** (Banos et al., 2014). This dataset contains data from 10 subjects for the task of human activity recognition, with 12 classes. The data was recorded with a 3-axis accelerometer, 3-axis gyroscope, and 3-axis magnetometer device at two different locations on the body, and a third device on the chest with a 3-axis accelerometer and two-lead ECG (total 23 channels), with a sampling frequency of 50 Hz. We segment the dataset with a window length of 100 samples (2 seconds).

**PAMAP** (Reiss & Stricker, 2012). This dataset contains data from 9 subjects for the task of human activity recognition, with 18 classes. The data was recorded with a temperature sensor, 3-axis accelerometer, 3-axis gyroscope, 3-axis magnetometer device at three different locations on the body (total 30 channels), with a sampling frequency of 100 Hz. To ease computation, we downsample the data from 100 Hz to 50 Hz. We segment the dataset with a window length of 150 samples (3 seconds).

**WISDM** (Kwapisz et al., 2011). This dataset contains data from 36 subjects for the task of human activity recognition, with six classes: walking, jogging, walking upstairs, walking downstairs, sitting, and standing. The data was recorded with a 3-axis accelerometer, with a sampling frequency of 20 Hz. We segment the dataset with a window length of 128 samples (6.4 seconds).

## B.2 Motor imagery datasets

**BNCI-1** (Tangermann et al., 2012). This dataset contains data from 9 subjects for the task of motor imagery classification, with 4 classes: left hand, right hand, feet, and tongue. The data was recorded with 22 electrodes, with a sampling frequency of 250 Hz. We filter the data with a bandpass filter with cutoff frequencies at 8 and 30 Hz, and we downsample the data to 100 Hz to ease computation. The data is already segmented into windows of 400 samples (4 seconds).

**BNCI-2** (Steyrl et al., 2016). This dataset contains data from 14 subjects for the task of motor imagery classification, with 2 classes: right hand and feet. The data was recorded with 15 electrodes, with a sampling frequency of 512 Hz. We filter the data with a bandpass filter with cutoff frequencies at 8 and 30 Hz, and we downsample the data to 100 Hz. The data is already segmented into windows of 500 samples (5 seconds).

**BNCI-4** (Leeb et al., 2007). This dataset contains data from 9 subjects for the task of motor imagery classification, with 2 classes: left hand and right hand. The data was recorded with 3 electrodes, with a sampling frequency of 250 Hz. We filter the data with a bandpass filter with cutoff frequencies at 8 and 30 Hz, and we downsample the data to 100 Hz. The data is already segmented into windows of 750 samples (7.5 seconds).

**ZHOU** (Zhou et al., 2016). This dataset contains data from 4 subjects for the task of motor imagery classification, with 3 classes: left hand, right hand, and feet. The data was recorded with 14 electrodes, with a sampling frequency of 250 Hz. We filter the data with a bandpass filter with cutoff frequencies at 8 and 30 Hz, and we downsample the data to 100 Hz. The data is already segmented into windows of 500 samples (5 seconds).

## C   Algorithms

**ERM** (Vapnik, 1991). The objective of ERM in domain generalisation is to learn a model that minimises the empirical risk across all source domains. For a hypothesis class $\mathcal{F}$, this is:

$$\arg\min_{f \in \mathcal{F}} \sum_{d \in \mathcal{E}_{\text{train}}} \alpha_d \hat{R}_d(f),$$

where $\alpha_d$ is the mixture weight for domain $d$ and $\hat{R}_d$ is the empirical risk on domain $d$:

$$\hat{R}_d(f) = \frac{1}{|D_d|} \sum_{(\mathbf{X}, y) \in D_d} \ell(f(\mathbf{X}), y).$$

**PhASER** (Mohapatra et al., 2025). The **PhASER** (Phase-Augmented Separate Encoding and Residual) framework consists of three components. The first is data augmentation with the Hilbert transform, which generates a $\pi/2$ phase-shifted version of each sample that is then merged with the original data. The second is representing the time series as magnitude and phase components using the short-time Fourier transform, and encoding these with separate encoders. The third is a phase-driven residual connection that broadcasts phase features to the temporal encoder output. The model is trained with standard ERM.

**Domain generalisation algorithms.** The following domain generalisation algorithms minimise a loss function comprised of the empirical risk and a penalty term:

$$\mathcal{L}(f) = \sum_{d \in \mathcal{E}_{\text{train}}} \alpha_d \hat{R}_d(f) \; + \; \lambda \mathcal{P}(f),$$

where $\mathcal{P}(f)$ is the algorithm-specific penalty, and $\lambda$ weights the penalty relative to the empirical risk.

**MMD** (Li et al., 2018b). The penalty is the domain divergence between pairs of source domains, measured with the maximum mean discrepancy:

$$\mathcal{P}_{\text{MMD}}(f) = \sum_{d < d'} \text{MMD}^2(\hat{P}_d^g(\mathbf{z}), \hat{P}_{d'}^g(\mathbf{z})),$$

where $\hat{P}_d^g(\mathbf{z}) = \left\{ \mathbf{z}_i = g(\mathbf{X}_i) \mid \mathbf{X}_i \in D_d \right\}$ is the empirical feature distribution for domain $d$ and the empirical estimate of the MMD is given by:

$$\text{MMD}^2(\hat{P}_d^g(\mathbf{z}), \hat{P}_{d'}^g(\mathbf{z})) = \frac{1}{|D_d|^2} \sum_{i,i'} k(\mathbf{z}_i, \mathbf{z}_{i'}) + \frac{1}{|D_{d'}|^2} \sum_{j,j'} k(\mathbf{z}_j, \mathbf{z}_{j'}) - \frac{2}{|D_d||D_{d'}|} \sum_{i,j} k(\mathbf{z}_i, \mathbf{z}_j).$$

We use the energy distance:

$$k(x, y) = -||x - y||_2,$$

and compute the loss using the GeomLoss library (Feydy et al., 2019).

**VREx** (Krueger et al., 2021). The penalty is the variance of the empirical risks on each source domain:

$$\mathcal{P}_{\text{VREx}}(f) = \text{Var}_{d \in \mathcal{E}_{\text{train}}} \left( \hat{R}_d(f) \right).$$

**NNR** (Shi et al., 2024). The penalty is the nuclear norm (the sum of the singular values) of the features from all source domains. Let $\mathbf{M} \in \mathbb{R}^{n \times m}$ be the matrix whose rows are the feature vectors from all source

domains, where $n = \sum_{d \in \mathcal{E}_{\text{train}}} |D_d|$ is the total number of feature vectors and $m$ is the feature dimension. The penalty is the nuclear norm of this feature matrix:

$$\mathcal{P}_{\text{NNR}}(f) = \|\mathbf{M}\|_* = \sum_{i=1}^{\min\{m,n\}} \sigma_i(\mathbf{M}),$$

where $\sigma_i(\mathbf{M})$ is the $i$th largest singular value.

**Setting $\lambda$.** A good value for $\lambda$ depends on the scale of the penalty term $\mathcal{P}(f)$, as it is balanced against the empirical risk $\hat{R}(f)$. If $\lambda$ is too large, the penalty dominates the loss and optimisation prioritises minimising $\mathcal{P}(f)$ at the expense of empirical risk. If $\lambda$ is too small, the algorithm effectively reduces to standard ERM. To select suitable ranges of $\lambda$ for each algorithm, we carried out preliminary experiments to identify when these adverse behaviours occur. Based on these experiments, we evaluate the following values:

- MMD: $\lambda \in [0.1, 0.5, 1, 2, 5]$

- VREx: $\lambda \in [1, 5, 10, 20, 50]$

- NNR: $\lambda \in [0.0001, 0.0005, 0.001, 0.002, 0.005]$

**Sampling strategy.** MMD and VREx have penalty terms computed on individual domains, so we use domain-balanced batches (as in DomainBed (Gulrajani & Lopez-Paz, 2021) and WOODS (Gagnon-Audet et al., 2023)) to ensure each domain has sufficient samples. ERM, PhASER, and NNR do not use domain labels, so we use standard uniform sampling from the combined training data.

## D   Model details

### D.1   Practical details for implementing middle and late fusion

The most straightforward way to implement middle and late fusion is to loop over individual channel-specific encoders or models. However, this becomes computationally expensive as $C$ grows, as it prevents parallel processing of channels. For the CNN, we use grouped convolutions with one group per channel, giving each channel private convolutional weights while allowing all channels to be processed in parallel. The LSTM and transformer do not provide an analogous grouping mechanism. For these, we use the looped approach, and manage computational costs by modifying the experiment design, as detailed in Appendix E.1.

### D.2   Model structures

**CNN.** The CNN encoder consists of 3 1D convolutional layers, each with 8 kernels of length 3, with normalisation and leaky ReLU activation after each layer. For early fusion we use batch normalisation, and for middle and late fusion we use group normalisation.

**CNN-LSTM.** The same convolutional structure as the CNN is used. The resulting representations (pre-pooling) are passed to a single bidirectional LSTM layer with hidden size 4 per direction, yielding an 8-dimensional representation.

**Transformer.** The input is projected to an 8-dimensional embedding space with a linear layer, and sinusoidal positional encodings are added. The sequence is processed by a 2-layer transformer encoder with 2 attention heads, feedforward dimension 16, and dropout 0.2.

Models use global average pooling along the temporal dimension of the final representation to obtain an 8-dimensional feature vector.

**PhASER.** The input time series is converted to amplitude and phase using the short-time Fourier transform, yielding two tensors of shape (channels × frequencies × time steps). All encoders (magnitude, phase, fusion, depth, temporal, phase residual) are implemented as 2D convolutional layers, with batch normalisation and SiLU activations.

Table 4: Full set of results.

| | DSADS | MHEALTH | PAMAP | WISDM | BNCI-1 | BNCI-2 | BNCI-4 | ZHOU |
|---|---|---|---|---|---|---|---|---|
| **Average** ($m_{\text{avg}}^{\text{ID}}$) ($\uparrow$) | | | | | | | | |
| early | $0.969_{\pm0.004}$ | $0.992_{\pm0.001}$ | $0.947_{\pm0.002}$ | $0.904_{\pm0.002}$ | $0.484_{\pm0.021}$ | $0.687_{\pm0.020}$ | $0.726_{\pm0.004}$ | $0.802_{\pm0.012}$ |
| middle | $0.969_{\pm0.002}$ | $0.993_{\pm0.002}$ | $0.945_{\pm0.003}$ | $0.903_{\pm0.005}$ | $0.201_{\pm0.010}$ | $0.397_{\pm0.021}$ | $0.544_{\pm0.019}$ | $0.627_{\pm0.035}$ |
| middle-e | $0.971_{\pm0.003}$ | $0.997_{\pm0.002}$ | $0.945_{\pm0.003}$ | $0.905_{\pm0.002}$ | / | / | / | / |
| middle-n | $0.958_{\pm0.004}$ | $0.991_{\pm0.001}$ | $0.934_{\pm0.004}$ | $\mathbf{0.915}_{\pm0.003}$ | / | / | / | / |
| middle-e-n | $\mathbf{0.972}_{\pm0.001}$ | $\mathbf{0.998}_{\pm0.001}$ | $0.942_{\pm0.003}$ | $0.910_{\pm0.003}$ | / | / | / | / |
| late | $0.949_{\pm0.003}$ | $0.993_{\pm0.001}$ | $0.912_{\pm0.004}$ | $0.860_{\pm0.007}$ | $0.196_{\pm0.011}$ | $0.376_{\pm0.013}$ | $0.545_{\pm0.026}$ | $0.464_{\pm0.017}$ |
| MMD | $0.968_{\pm0.005}$ | $0.996_{\pm0.001}$ | $0.943_{\pm0.002}$ | $0.121_{\pm0.003}$ | $0.433_{\pm0.016}$ | $0.517_{\pm0.040}$ | $0.709_{\pm0.012}$ | $0.799_{\pm0.017}$ |
| VREx | $0.969_{\pm0.003}$ | $0.996_{\pm0.001}$ | $0.944_{\pm0.001}$ | $0.115_{\pm0.005}$ | $0.444_{\pm0.020}$ | $0.619_{\pm0.016}$ | $0.714_{\pm0.008}$ | $0.793_{\pm0.020}$ |
| NNR | $0.968_{\pm0.002}$ | $0.994_{\pm0.000}$ | $\mathbf{0.947}_{\pm0.003}$ | $0.905_{\pm0.002}$ | $\mathbf{0.485}_{\pm0.013}$ | $\mathbf{0.697}_{\pm0.011}$ | $0.723_{\pm0.007}$ | $\mathbf{0.816}_{\pm0.015}$ |
| PhASER | $0.950_{\pm0.007}$ | $0.980_{\pm0.002}$ | $0.921_{\pm0.005}$ | $0.896_{\pm0.002}$ | $0.296_{\pm0.026}$ | $0.587_{\pm0.023}$ | $\mathbf{0.736}_{\pm0.010}$ | $0.641_{\pm0.044}$ |
| **Average** ($m_{\text{avg}}^{\text{OOD}}$) ($\uparrow$) | | | | | | | | |
| early | $0.733_{\pm0.021}$ | $0.873_{\pm0.015}$ | $0.770_{\pm0.016}$ | $0.739_{\pm0.011}$ | $0.376_{\pm0.014}$ | $0.590_{\pm0.020}$ | $0.698_{\pm0.007}$ | $0.701_{\pm0.030}$ |
| middle | $0.763_{\pm0.007}$ | $0.930_{\pm0.008}$ | $0.850_{\pm0.007}$ | $0.775_{\pm0.009}$ | $0.163_{\pm0.014}$ | $0.361_{\pm0.013}$ | $0.459_{\pm0.025}$ | $0.476_{\pm0.010}$ |
| middle-e | $0.748_{\pm0.009}$ | $0.915_{\pm0.020}$ | $0.856_{\pm0.003}$ | $0.772_{\pm0.007}$ | / | / | / | / |
| middle-n | $0.806_{\pm0.006}$ | $0.937_{\pm0.004}$ | $0.868_{\pm0.010}$ | $\mathbf{0.792}_{\pm0.014}$ | / | / | / | / |
| middle-e-n | $0.814_{\pm0.015}$ | $0.943_{\pm0.010}$ | $\mathbf{0.894}_{\pm0.008}$ | $0.768_{\pm0.009}$ | / | / | / | / |
| late | $\mathbf{0.849}_{\pm0.012}$ | $\mathbf{0.945}_{\pm0.002}$ | $0.865_{\pm0.006}$ | $0.742_{\pm0.015}$ | $0.154_{\pm0.012}$ | $0.354_{\pm0.017}$ | $0.472_{\pm0.024}$ | $0.356_{\pm0.020}$ |
| MMD | $0.685_{\pm0.029}$ | $0.836_{\pm0.014}$ | $0.762_{\pm0.014}$ | $0.117_{\pm0.008}$ | $0.355_{\pm0.012}$ | $0.478_{\pm0.037}$ | $0.686_{\pm0.011}$ | $0.705_{\pm0.017}$ |
| VREx | $0.704_{\pm0.020}$ | $0.835_{\pm0.023}$ | $0.758_{\pm0.010}$ | $0.112_{\pm0.009}$ | $0.383_{\pm0.008}$ | $0.542_{\pm0.026}$ | $0.691_{\pm0.010}$ | $0.701_{\pm0.035}$ |
| NNR | $0.741_{\pm0.027}$ | $0.877_{\pm0.011}$ | $0.770_{\pm0.015}$ | $0.742_{\pm0.015}$ | $\mathbf{0.398}_{\pm0.015}$ | $\mathbf{0.611}_{\pm0.013}$ | $0.694_{\pm0.012}$ | $\mathbf{0.720}_{\pm0.036}$ |
| PhASER | $0.749_{\pm0.013}$ | $0.859_{\pm0.007}$ | $0.822_{\pm0.010}$ | $0.781_{\pm0.009}$ | $0.259_{\pm0.015}$ | $0.545_{\pm0.022}$ | $\mathbf{0.699}_{\pm0.014}$ | $0.595_{\pm0.041}$ |
| **Worst-group** ($m_{\text{worst}}^{\text{OOD}}$) ($\uparrow$) | | | | | | | | |
| early | $0.456_{\pm0.039}$ | $0.759_{\pm0.043}$ | $0.419_{\pm0.045}$ | $0.118_{\pm0.065}$ | $0.150_{\pm0.019}$ | $0.333_{\pm0.000}$ | $0.486_{\pm0.044}$ | $0.590_{\pm0.082}$ |
| middle | $0.597_{\pm0.052}$ | $0.825_{\pm0.019}$ | $0.716_{\pm0.052}$ | $0.221_{\pm0.092}$ | $0.100_{\pm0.000}$ | $0.331_{\pm0.003}$ | $0.341_{\pm0.022}$ | $0.261_{\pm0.038}$ |
| middle-e | $0.603_{\pm0.055}$ | $0.786_{\pm0.037}$ | $0.711_{\pm0.018}$ | $0.196_{\pm0.053}$ | / | / | / | / |
| middle-n | $0.720_{\pm0.018}$ | $0.819_{\pm0.020}$ | $0.769_{\pm0.043}$ | $0.350_{\pm0.043}$ | / | / | / | / |
| middle-e-n | $0.720_{\pm0.030}$ | $\mathbf{0.865}_{\pm0.026}$ | $\mathbf{0.816}_{\pm0.017}$ | $0.305_{\pm0.026}$ | / | / | / | / |
| late | $\mathbf{0.770}_{\pm0.023}$ | $0.858_{\pm0.024}$ | $0.778_{\pm0.024}$ | $0.350_{\pm0.039}$ | $0.101_{\pm0.003}$ | $0.333_{\pm0.001}$ | $0.344_{\pm0.026}$ | $0.242_{\pm0.036}$ |
| MMD | $0.476_{\pm0.041}$ | $0.644_{\pm0.051}$ | $0.467_{\pm0.035}$ | $0.004_{\pm0.003}$ | $0.168_{\pm0.039}$ | $0.332_{\pm0.001}$ | $0.529_{\pm0.011}$ | $0.604_{\pm0.020}$ |
| VREx | $0.520_{\pm0.020}$ | $0.681_{\pm0.063}$ | $0.412_{\pm0.057}$ | $0.005_{\pm0.003}$ | $0.172_{\pm0.014}$ | $0.333_{\pm0.001}$ | $0.522_{\pm0.022}$ | $0.592_{\pm0.086}$ |
| NNR | $0.538_{\pm0.016}$ | $0.699_{\pm0.020}$ | $0.441_{\pm0.074}$ | $0.207_{\pm0.127}$ | $\mathbf{0.177}_{\pm0.031}$ | $0.333_{\pm0.000}$ | $0.513_{\pm0.032}$ | $\mathbf{0.609}_{\pm0.056}$ |
| PhASER | $0.553_{\pm0.038}$ | $0.737_{\pm0.023}$ | $0.719_{\pm0.030}$ | $\mathbf{0.361}_{\pm0.031}$ | $0.155_{\pm0.020}$ | $\mathbf{0.348}_{\pm0.030}$ | $\mathbf{0.541}_{\pm0.023}$ | $0.475_{\pm0.072}$ |
| **Gap** ($\text{gap}_{\text{avg}}$) ($\downarrow$) | | | | | | | | |
| early | $0.236_{\pm0.019}$ | $0.118_{\pm0.015}$ | $0.177_{\pm0.016}$ | $0.165_{\pm0.011}$ | $0.109_{\pm0.020}$ | $0.096_{\pm0.010}$ | $0.028_{\pm0.008}$ | $0.102_{\pm0.035}$ |
| middle | $0.206_{\pm0.007}$ | $0.064_{\pm0.007}$ | $0.095_{\pm0.008}$ | $0.128_{\pm0.009}$ | $0.038_{\pm0.020}$ | $0.036_{\pm0.013}$ | $0.085_{\pm0.010}$ | $0.151_{\pm0.033}$ |
| middle-e | $0.223_{\pm0.007}$ | $0.082_{\pm0.018}$ | $0.089_{\pm0.004}$ | $0.133_{\pm0.008}$ | / | / | / | / |
| middle-n | $0.152_{\pm0.006}$ | $0.054_{\pm0.003}$ | $0.066_{\pm0.010}$ | $0.123_{\pm0.015}$ | / | / | / | / |
| middle-e-n | $0.158_{\pm0.015}$ | $0.055_{\pm0.010}$ | $0.048_{\pm0.007}$ | $0.143_{\pm0.009}$ | / | / | / | / |
| late | $\mathbf{0.100}_{\pm0.012}$ | $\mathbf{0.048}_{\pm0.002}$ | $\mathbf{0.047}_{\pm0.003}$ | $0.117_{\pm0.009}$ | $0.042_{\pm0.014}$ | $\mathbf{0.022}_{\pm0.026}$ | $0.073_{\pm0.014}$ | $0.107_{\pm0.028}$ |
| MMD | $0.283_{\pm0.026}$ | $0.161_{\pm0.014}$ | $0.181_{\pm0.013}$ | $0.004_{\pm0.006}$ | $0.078_{\pm0.015}$ | $0.040_{\pm0.033}$ | $0.024_{\pm0.004}$ | $0.094_{\pm0.023}$ |
| VREx | $0.265_{\pm0.018}$ | $0.161_{\pm0.022}$ | $0.186_{\pm0.010}$ | $\mathbf{0.003}_{\pm0.004}$ | $0.061_{\pm0.013}$ | $0.077_{\pm0.027}$ | $\mathbf{0.023}_{\pm0.003}$ | $0.092_{\pm0.024}$ |
| NNR | $0.227_{\pm0.025}$ | $0.117_{\pm0.011}$ | $0.177_{\pm0.014}$ | $0.163_{\pm0.016}$ | $0.087_{\pm0.014}$ | $0.086_{\pm0.018}$ | $0.028_{\pm0.007}$ | $0.096_{\pm0.026}$ |
| PhASER | $0.201_{\pm0.013}$ | $0.121_{\pm0.006}$ | $0.099_{\pm0.009}$ | $0.115_{\pm0.008}$ | $\mathbf{0.037}_{\pm0.013}$ | $0.042_{\pm0.024}$ | $0.037_{\pm0.006}$ | $\mathbf{0.045}_{\pm0.019}$ |

**Training.** All models are trained with the Adam optimiser with a learning rate of $5 \times 10^{-3}$ and weight decay $1 \times 10^{-5}$, with batch size of 64. Models are trained with early stopping based on the cross-entropy loss on the validation set.

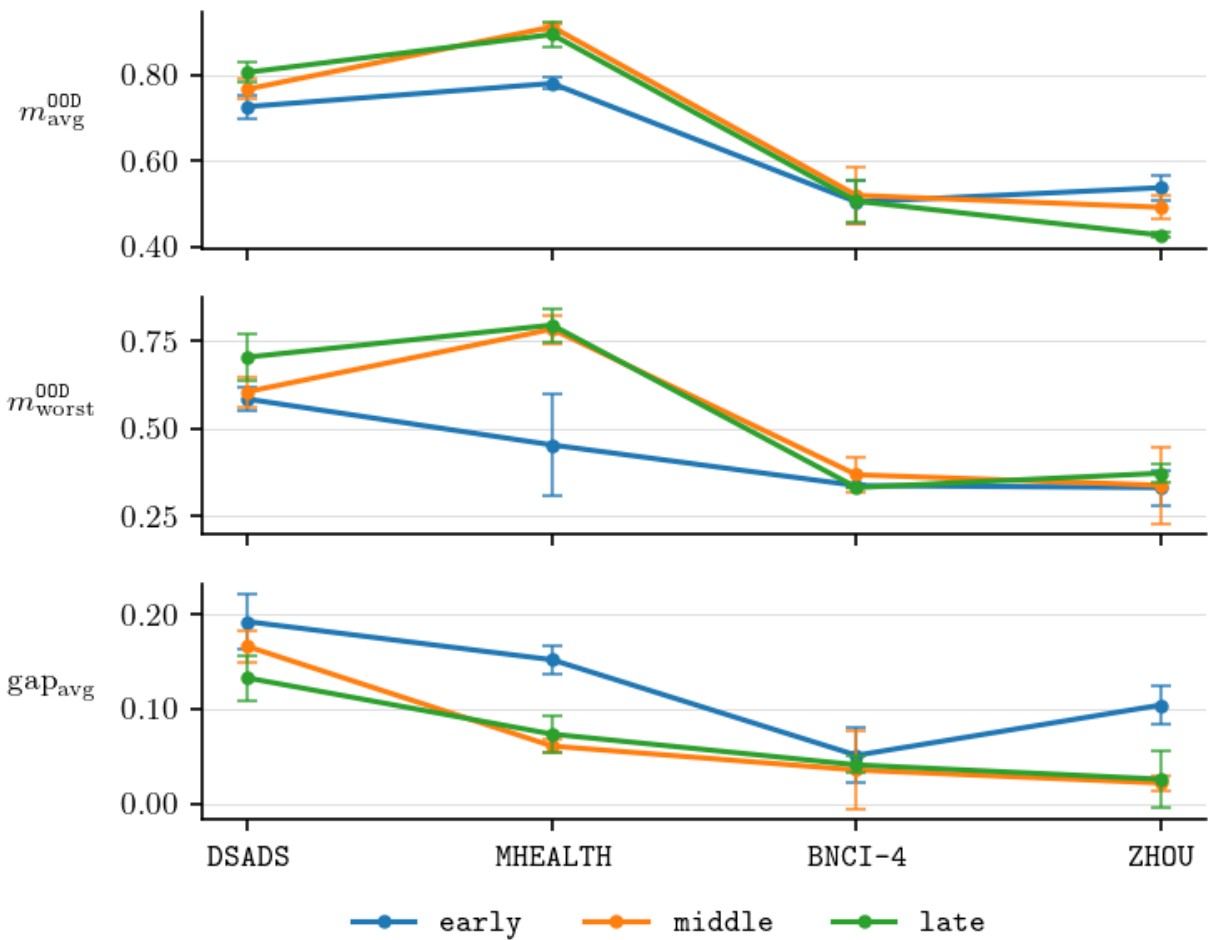

Figure 10: Average OOD performance $m_{\mathrm{avg}}^{\mathrm{OOD}}$, worst-group performance $m_{\mathrm{worst}}^{\mathrm{OOD}}$, and generalisation gap $\mathrm{gap}_{\mathrm{avg}}$ across datasets for early, middle, late with a CNN-LSTM encoder.

# E    Full results

## E.1    Results with different encoders

The main experiments use a 1D CNN as the encoder. In this experiment, we study OOD performance with two other encoders: CNN-LSTM and transformer. As mentioned in Appendix D.1, we modify these experiments to manage computational costs. Specifically, we use three experiment seeds (instead of five), four datasets (DSADS, MHEALTH, BNCI4, ZHOU), and randomly subsample the channels for each seed (randomly select $\lfloor C/2 \rfloor$ channels).

Figures 10 and 11 show the OOD performance for early, middle, and late, with a CNN-LSTM and transformer encoder, respectively. The same patterns described in Section 4.4 hold across both encoders, indicating that our findings are not specific to a 1D CNN encoder.

## E.2    Parameter count experiments

In this experiment, we examine whether the larger number of parameters of middle and late models are responsible for improved OOD performance on the HAR datasets.

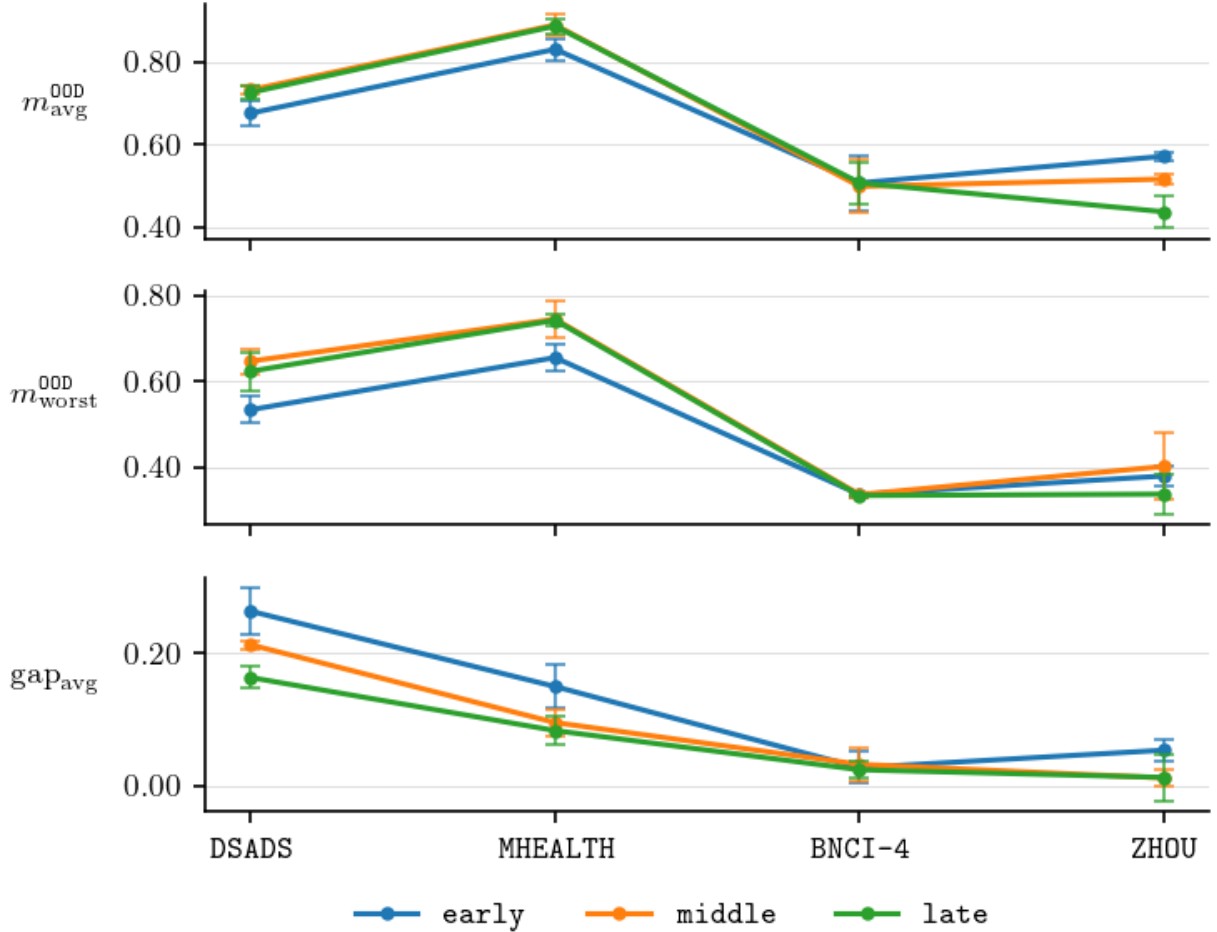

Figure 11: Average OOD performance $m_{\mathrm{avg}}^{\mathrm{OOD}}$, worst-group performance $m_{\mathrm{worst}}^{\mathrm{OOD}}$, and generalisation gap $\mathrm{gap}_{\mathrm{avg}}$ across datasets for `early`, `middle`, `late` with a transformer encoder.

Table 5: Number of trainable parameters for `early`, `middle`, and `late` models. $n_f$ is the number of filters in each layer of the 1D CNN encoder(s).

| | $n_f$ | DSADS | MHEALTH | PAMAP | WISDM |
|---|---|---|---|---|---|
| early | 8 | 963 | 1,092 | 1,260 | 540 |
| early | 64 | 29,075 | 30,156 | 31,500 | 25,796 |
| middle | 8 | 7,026 | 10,619 | 13,818 | 1,407 |
| late | 8 | 9,405 | 12,972 | 16,920 | 1,476 |

Figure 12 shows the OOD performance for `early` with $n_f = 8$ and $n_f = 64$, and Table 5 shows the parameter counts. While increasing parameter count improves `early` performance, `middle` and `late` with $n_f = 8$ still largely outperform `early` with $n_f = 64$ despite having fewer parameters.

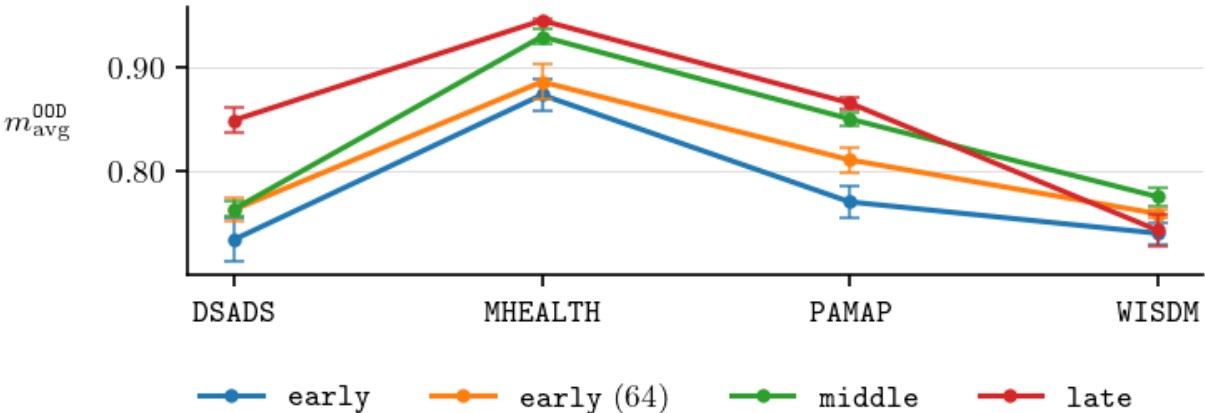

Figure 12: Average OOD performance $m_{\text{avg}}^{\text{OOD}}$ across the HAR datasets. The number in the bracket is $n_f$, the number of filters in each layer of the 1D CNN encoder.

