# OpenReview forum: "Rethinking channel fusion for robust multivariate time series classification under distribution shift"
_TMLR — Under review for TMLR_

### Review · Reviewer_MNPe · 2026-06-19

**Summary Of Contributions:**

The paper explores the impact of channel fusion strategies on OOD generalization in multivariate time series classification. The authors show both theoretically and empirically that later fusion strategies structurally isolate distribution shifts, preventing localized corruptions from contaminating the entire feature representation. Through extensive evaluation on multiple datasets, the authors show a data dependent capacity & robustness trade-off. Lastly, the authors propose a practical and heuristic methods to automatically select the optimal fusion strategy without target domain data.

**Additional Comments:**

Please see my weakness points above

**Audience:**

Yes

**Audience Explanation:**

The authors propose a simple and practical ID-based heuristic for selecting the optimal fusion strategy without needing access to target domain data. Also, the authors provide useful architectural ablations which identify L2-normalization and uniform weighting as optimal design choices for later fusion. Those empirical findings are beneficial for the TMLR audience who are interested in multi-variate time-series classification problems.

**Claims And Evidence:**

Yes

**Claims Explanation:**

The authors clearly highlights a critical architectural design choice that significantly impacts OOD robustness. Also, the paper provides a solid analysis of how distribution shifts and localized perturbations propagate via different fusion architectures. Those claims are supported by empirical visualizations. Also, the authors provided extensive empirical evaluations across multiple datasets with various settings.

**Requested Changes:**

**Weakness of the paper:**
- Empirical evaluation excludes some highly relevant baselines. For example, the authors cite DIVERSIFY but not include it in the experiments. Some other temporal covariate shift and DG baselines like GroupDRO, IRM and AdaRNN are also missing.
- The paper somewhat relied on custom leave-one-out splits on specific datasets instead of evaluating on standardized time-series OOD datasets (e.g., WOODS benchmark) even though authors cite in the introduction.
- The computational complexity of late fusion is not well addressed in the manuscript. Since late fusion requires training independent encoders and classifiers for every channel, it scales linearly. This could be less practical for high-density sensor arrays or sequence models without grouped convolutions.

---

### Review · Reviewer_HFpi · 2026-07-14

**Summary Of Contributions:**

This work focuses on the question: how does the stage at which sensor channels are fused affect multivariate time-series classification under distribution shift?

they compare:

- Early fusion using a shared encoder over all channels.
- Middle fusion using channel-specific encoders followed by weighted feature aggregation and a shared classifier.
- Late fusion using independently trained channel-specific models whose predictions are combined.

The comparison covers four human-activity-recognition (HAR) datasets and four motor-imagery EEG datasets under leave-one-subject-out evaluation. They also combine the architectures with several domain-generalization methods and study synthetic channel corruption on DSADS. Finally, they propose an ID-performance-tolerance heuristic for deciding when a later-fusion model should replace early fusion.

Several aspects of the study are valuable. The early/middle/late distinction is presented clearly. The equations showing how a perturbation can propagate through early versus late fusion provide intuition. The paper reports not only average OOD performance but also ID performance, worst-subject performance, and the ID–OOD gap. They report a negative result on motor-imagery data rather than suggesting that late fusion is universally preferable. The comparison between architectural choices and established domain-generalization objectives is also good.

The central empirical observation is that later fusion performs favorably on the examined HAR datasets, particularly under synthetic channel corruption but suffers a substantial loss of absolute performance on the motor-imagery datasets. This is an interesting and potentially publishable result. However, there are several experimental confounds and they prevent attributing the differences cleanly to fusion stage alone.

**Additional Comments:**

NA

**Audience:**

Yes

**Audience Explanation:**

fusion architecture is a practically important and comparatively under-controlled design choice in multivariate time-series learning.

The contrast between HAR and motor-imagery results, including the failure of later fusion in the latter setting, is scientifically informative.

The paper also fits TMLR’s emphasis on careful technical investigation even if the conclusions are not a universally superior new method.

The paper would be relevant to researchers working on multivariate time series, sensor robustness, domain generalization, wearable computing, and EEG classification.

**Claims And Evidence:**

No

**Claims Explanation:**

1. Appendix D.2 states that the early-fusion CNN uses batch normalization, whereas the middle- and late-fusion models use group normalization. Normalization can affect behavior under subject shift, small per-domain batches, and corrupted inputs. Consequently, the comparison does not isolate channel fusion. All architectures should be compared under the same normalization scheme or a complete normalization-by-fusion ablation should be reported.

2. A late-fusion model contains multiple independently trained predictors. Its robustness may therefore arise partly from ensembling and prediction diversity rather than channel isolation. Increasing the width of a single early-fusion model, as done in the appendix, does not control for this effect. A matched early-fusion ensemble, with comparable member count, parameter budget, training compute, and inference cost, would help separate the ensemble effect from the channel-fusion effect.

3. The theoretical presentation needs correction. As written the domain-adaptation bound appears to conflate the generic H-divergence with the H ΔH-divergence used in the standard binary-classification bound. the factor of 1/2 also depends on the stated definition. The loss and hypothesis-class assumptions should be explicit.

4. More substantively, covariate shift does not imply that the joint optimal risk is negligible. There may be irreducible label noise or hypothesis-class approximation error. Moreover, after a learned, potentially non-invertible representation g(X), equality of P(y|X) does not necessarily imply equality of P(y|g(X)). These qualifications matter because the argument is applied at the representation level.

5.. The empirical divergence measure does not instantiate the stated bound. The experiments replace the bound’s hypothesis-class divergence with Wasserstein distance in learned feature spaces. Those feature spaces can differ in dimensionality, scale, and geometry between fusion architectures. Similar average feature magnitudes do not establish that their Wasserstein estimates are comparable, and finite-sample high-dimensional Wasserstein estimates can be unstable. A domain-classifier-based proxy distance or a normalized kernel discrepancy, together with uncertainty estimates, would make the analysis more convincing. Otherwise the result should be framed as an exploratory association rather than validation of the bound.

6. The proposed selection rule is evaluated post hoc using target performance. Once delta is fixed, the rule uses only ID performance. However, the recommended range of delta is inferred by examining OOD results on these same datasets. Therefore, the current experiments do not yet demonstrate target-free selection of delta. A nested evaluation (for example, selecting delta on other datasets or held-out domains and evaluating it on an unseen dataset) would be needed to support the deployment claim.

7. The corruption experiment may structurally favor later fusion. “Important” channels are selected using the gradients of the early-fusion model and then corrupted for every architecture. A channel important to early fusion need not be equally important to middle or late fusion. Model-specific importance estimates, model-agnostic sensor groups, and random or exhaustive channel subsets should be reported separately.

8. The scope of the corruption evidence is limited. The corruption study is conducted only on DSADS and appears to corrupt individual scalar axes. DSADS’s 15 retained accelerometer channels correspond to five tri-axial sensors, so corrupting seven axes is not equivalent to the failure of half the sensors. Real device failure would commonly affect all axes from a device. The manuscript should either use grouped, sensor-level failures or consistently describe this as axis/channel corruption. Claims about robustness to sensor failure should also be limited to the tested dataset unless confirmed elsewhere.

9. A small ID–OOD gap is not evidence of robustness. The gap can be reduced simply by lowering ID performance, which is approximately what occurs for later fusion on the motor-imagery datasets. Absolute OOD performance and performance retention should remain primary. The paper recognizes this issue in places but some text of the paper still treats gap reduction as a robustness benefit.

10.  Higher average cross-channel Pearson correlation in EEG does not establish that useful cross-channel interactions cause early fusion’s advantage. EEG correlation can also reflect volume conduction, common referencing, artifacts, or non-discriminative shared activity. A stronger test would disrupt cross-channel alignment while preserving marginal channel signals, or compare class-conditional interaction measures. Until then, the proposed mechanism should be presented as a hypothesis.

**Requested Changes:**

The requested changes can be found at the previous part of the review, in which I enumerated 10 technical weaknesses in my opinion.

Additionally, some changes in terms of positioning relative to prior work:

Compare more directly with relevant work on channel-specific and corruption-robust fusion, including:

Xaviar et al., Robust Multimodal Fusion for Human Activity Recognition, which studies early, late, and multilevel fusion under noisy or missing sensor modalities.

Ek et al., Transformer-based Models to Deal with Heterogeneous Environments in Human Activity Recognition, which introduces sensor-oriented processing for robustness to heterogeneous devices and placements.

Chandankar and Burchard, Sensor-Specific Transformer Ensembles with Test-Matched Augmentation, which is especially close to independent per-sensor modeling followed by probability-level fusion.

Zucchi and Lampert, PRISM: A Structural Regularization Approach for Channel-Independent Multivariate Time-Series Classification, which provides a related channel-independent formulation.

The already-cited SSSS-TSA work also merits a more explicit methodological comparison.

In light of this literature, the novelty claim should emphasize the systematic cross-task and domain-shift comparison rather than suggesting that the robustness implications of fusion or channel separation are broadly unexplored.

---

### Review · Reviewer_WSm7 · 2026-07-15

**Summary Of Contributions:**

This paper studies the role of channel fusion strategies in robust multivariate time series classification under distribution shifts. Unlike existing works that mainly focus on domain generalization algorithms, the authors show that the choice of fusion architecture itself can significantly affect OOD robustness.

The paper compares early, middle, and late fusion strategies, and provides an analysis of how channel-specific perturbations propagate through different architectures. Experiments on four HAR datasets and four MI datasets demonstrate that later fusion can substantially improve robustness under subject-level shifts and sensor corruptions, especially for HAR tasks, while early fusion remains preferable for MI tasks where cross-channel interactions are important. The authors also investigate several design choices within later fusion and propose a simple fusion selection heuristic based on ID validation performance.

The paper provides an interesting perspective that architectural design choices can serve as robustness inductive biases. The main limitations are that the experiments are mainly limited to sensor-based datasets, and the theoretical analysis provides intuition rather than formal guarantees.

**Additional Comments:**

This paper presents a useful empirical investigation of an overlooked factor in multivariate time series robustness. The main contribution is the observation that channel fusion strategy can substantially influence OOD generalization, providing a complementary perspective to existing domain generalization methods.

The paper is well motivated and experimentally solid. While the conclusions are currently supported mainly by sensor-based benchmarks and the theoretical analysis is primarily explanatory, I believe the work provides valuable insights and would be of interest to the TMLR community. I lean towards acceptance.

**Audience:**

Yes

**Audience Explanation:**

Channel fusion is a fundamental design choice in multivariate time series models, but its impact on OOD robustness has received limited attention. This work provides a useful empirical study showing that model architecture can be as important as the choice of training algorithms for robustness. The findings should be relevant to researchers working on time series classification and domain generalization.

**Broader Impact Concerns:**

I do not find significant broader impact concerns. The work focuses on improving robustness of time series models and does not introduce additional ethical risks. For applications involving sensitive data such as EEG or healthcare, standard considerations regarding privacy and reliability still apply.

**Claims And Evidence:**

Yes

**Claims Explanation:**

The main claims are generally supported by the experimental results. The authors conduct evaluations under multiple types of distribution shifts and compare fusion strategies with several domain generalization baselines. The consistent improvements of later fusion on HAR datasets, together with the sensor corruption experiments, provide evidence for the claim that channel isolation can improve robustness.

However, the conclusions should be interpreted with appropriate scope. The results also show that later fusion is not universally better, as early fusion performs better on MI datasets due to stronger cross-channel dependencies. The theoretical analysis explains the observed behavior but does not provide formal guarantees.

**Requested Changes:**

- Clarify the scope of the conclusions. The paper should emphasize that later fusion is not universally preferable. Its advantages mainly come from isolating channel-specific shifts, while early fusion can be beneficial when cross-channel interactions contain important discriminative information.
- Discuss computational costs of late fusion. Late fusion requires independent channel-specific models, which may introduce additional training and inference costs. A brief discussion of these practical considerations would improve the paper.
- Improve the analysis of channel dependency. The current analysis mainly relies on channel correlation to explain the difference between HAR and MI datasets. Since correlation does not necessarily represent task-relevant dependency, further discussion of this limitation would strengthen the analysis.
- Improve figure consistency. The figures are informative, but the visual styles are somewhat inconsistent across the paper. A unified presentation style would improve readability.